# Sea-ice feedbacks influence the isotopic signature of Greenland Ice Sheet elevation changes: Last Interglacial HadCM3 simulations

Irene Malmierca-Vallet[1,2], Louise C. Sime[1], Paul J. Valdes[2], and Julia C. Tindall[3]

[1]British Antarctic Survey, High Cross, Madingley Road, Cambridge, CB3 0ET, UK
[2]School of Geographical Sciences, University of Bristol, University Road, Bristol, BS8 1SS, UK
[3]School of Earth and Environment, University of Leeds, Leeds, LS2 9JT, UK

**Correspondence:** I. Malmierca-Vallet (irealm37@bas.ac.uk)

**Abstract.**

Changes in the Greenland ice sheet (GIS) affect global sea level. Greenland stable water isotope ($\delta^{18}O$) records from ice cores offer information on past changes in the surface of the GIS. Here, we use the isotope-enabled HadCM3 climate model to simulate a set of Last Interglacial (LIG) idealised GIS surface elevation change scenarios focusing on GIS ice core sites. We investigate how $\delta^{18}O$ depends on the magnitude and sign of GIS elevation change and evaluate how the response is altered by sea ice changes. We find that modifying GIS elevation induces changes in Northern Hemisphere atmospheric circulation, sea ice and precipitation patterns. These climate feedbacks lead to ice core-averaged isotopic lapse rates of 0.49‰ per 100 m for the lowered GIS states and 0.29‰ per 100 m for the enlarged GIS states. This is lower than the spatially derived Greenland lapse rates of 0.62-0.72 ‰ per 100 m. These results thus suggest non-linearities in the isotope-elevation relationship, and have consequences for the interpretation of past elevation and climate changes across Greenland. In particular, our results suggest that winter sea ice changes may significantly influence isotopic-elevation gradients: winter sea ice effect can decrease (increase) modelled core-averaged isotopic lapse rate values by about -19% (and +28%) for the lowered (enlarged) GIS states respectively. The largest influence of sea ice on $\delta^{18}O$ changes is found in coastal regions like the Camp Century site.

## 1 Introduction

Ice core records of stable water isotopes ($\delta^{18}O$) yield useful information on past climate change over the last several glacial-interglacial cycles (e.g., Sime et al., 2009). Alongside site elevation, ice core $\delta^{18}O$ is affected by variations in site temperature, sea ice, evaporation conditions, and transport pathway effects (Sime et al., 2013; Werner et al., 2018). Improving our understanding of the elevation signal captured in Greenland ice core isotopic records means we also need to consider these impacts.

The Last Interglacial period (LIG - between around 130,000 and 115,000 ka) was the last time when the volume of the GIS is believed to have been considerably reduced (Robinson et al., 2011; Stone et al., 2013). This period was characterised by warmer-than-present-day conditions in the high latitudes and stronger summer time insolation (Hoffman et al., 2017; Capron et al., 2017). Over Arctic land areas, LIG summer temperatures are estimated to have been around 4-5°C above present-day (e.g., CAPE Last Interglacial Project Members, 2006) and the NEEM ice core record suggests surface temperatures 8 ± 4°C

warmer compared to the last millennium (when accounting for elevation changes in the GIS) (NEEM community members, 2013).

LIG global mean sea level is believed to have been risen by between 6 to 9 m compared to present-day levels (Kopp et al., 2009; Dutton et al., 2015) which likely indicates both reduced Antarctic (DeConto and Pollard, 2016; Sutter et al., 2016) and

5 Greenland ice sheets (Tarasov and Peltier, 2003; Lhomme et al., 2005; Helsen et al., 2013; Calov et al., 2015). The contribution of the GIS to this LIG high stand remains unclear: previous studies suggest a possible contribution anywhere between +0.3 to +5.5 m to global mean sea level (Cuffey and Marshall, 2000; Robinson et al., 2011; Quiquet et al., 2013; Stone et al., 2013; Plach et al., 2019). Interestingly though, some total air content measurements have been interpreted as indicative of the elevation over central Greenland remaining nearly unchanged (only few hundred meters lower than today) (Raynaud et al.,

1997) and the NEEM air content data have also been interpreted as indicative of a lowering of the surface elevation of only 130 ± 300 m relative to present (NEEM community members, 2013).

Seven deep ice cores, that likely contain some LIG ice, have been recovered from the GIS with stable water isotope records ($\delta^{18}O$ and $\delta D$): NEEM, NGRIP, GISP2, GRIP, DYE3, Camp Century and Renland (Johnsen and Vinther, 2007; NEEM community members, 2013). Isotope-elevation slopes derived from spatial data from central and northwest Greenland suggest that a

15 change in elevation of 100 m may provide a 0.62‰ and 0.72‰ change in $\delta^{18}O$ respectively (Dansgaard, 1973; Johnsen et al., 1989; Poage and Chamberlain, 2001). Though the suggested global average isotope lapse rate is 0.3‰ per 100m (Blisniuk and Stern, 2005), it is possible that isotope-elevation relationships vary more widely at high latitudes because of the higher temperature variability (Rowley and Garzione, 2007) or sea ice effects (Holloway et al., 2016a, 2017; Malmierca-Vallet et al., 2018).

While the LIG global average warming is in line with projections for the end of the century (Clark and Huybers, 2009; Hoffman et al., 2017), there is considerable uncertainty on the timing of the sea level high stand during the LIG (Kopp et al., 2009; Düsterhus et al., 2016; Barlow et al., 2018) and the magnitude considerably surpasses near (2100 - 2200) future projections (e.g., Fischer et al., 2018; Irvali et al., 2020). Thus, the LIG represents a relevant period, when the implications of changes in ice sheet elevation are highly pertinent for mid to far future projections (DeConto and Pollard, 2016). An improved understanding

of the isotopic response to GIS elevation changes may therefore help improve the interpretation of LIG Greenland isotope data and help constrain the GIS response to future sea level and temperature scenarios.

In this study, we first investigate the isolated impact of GIS elevation changes on Greenland $\delta^{18}O$ and the underlying processes (section 3.1). We perform a suite of idealised elevation change simulations with the isotope-enabled climate model HadCM3 to analyse the response of Greenland temperature (section 3.1.2) and precipitation (section 3.1.4), Arctic sea ice

(section 3.1.5) and atmospheric circulation (section 3.1.3) to these GIS elevation changes. The second part of this study focus on testing to which extent variations in the background climate state (Arctic sea ice extent) may influence the isotopic lapse rate values at different Greenland ice core sites (section 3.2). For this analysis, we additional use a second set of simulations that investigate the joint impact of Arctic sea ice change and GIS changes.

## 2   Methods

### 2.1   Model description

We use the isotope-enabled General Circulation Model (GCM) HadCM3 to simulate the isotopic response to idealised variations in the elevation of the GIS. This GCM has been widely used to examine present, past and future climates (Tindall et al., 2009, 2010; Sime et al., 2013; Holloway et al., 2016b) and consists of a coupled ocean, atmosphere and sea ice model. The atmosphere component has a horizontal grid spacing of 2.5° (latitude) by 3.75° (longitude) and has 19 vertical levels (Gordon et al., 2000). The horizontal grid resolution of the ocean component is 1.25° by 1.25° with 20 vertical levels (Gordon et al., 2000). Tindall et al. (2009) presents the implementation of the water isotope code in HadCM3.

The HadCM3 sea ice output over the Arctic Ocean has been previously validated against observational sea ice data (Meier et al., 2017; Peng et al., 2013) by Malmierca-Vallet et al. (2018). Under PI conditions, HadCM3 simulates rather little summer sea ice over the central Arctic Ocean, and too much winter sea ice over the Norwegian, Barents, Labrador and Bering Seas. For a full validation of the sea ice model, interested readers are referred to Malmierca-Vallet et al. (2018) and Gordon et al. (2000).

### 2.2   Experimental setup

#### 2.2.1   GIS elevation change simulations

We run a first ensemble of 16 idealised elevation change HadCM3 simulations with greenhouse-gas and orbital forcing centred at 125,000 years BP (125ka) (See Table A1). A 125ka control experiment (hereafter, 125Control) is performed including a present-day GIS configuration (IceBridge BedMachine Greenland, Version 3 – Morlighem et al. (2017a, b)). To generate the idealised elevation changes, we scale up and down the GIS height. In particular, we scale elevations relative to the elevation at the NEEM ice core site, following

(1)     $\beta = \Delta z / Z_{NEEM}$

where $\Delta z$ is the elevation change prescribed; $Z_{NEEM}$ is the elevation at the NEEM ice core site in the present-day GIS configuration and $\beta$ is the scaling percentage. GIS elevations are then decreased/increased by $\beta$ (ranging between ±2% to ±48%, Table. A1);

(2)     $Z_{new} = Z_{ini} \pm (\beta \ast Z_{ini})$

where $Z_{ini}$ is the two-dimension array for the present-day GIS and $Z_{new}$ is a new two-dimension array with modified elevations. Our simulations use $\Delta z$ equal to ± 50, 100, 300, 500, 700, 900, 1000, 1300 m (see Table A1). The 16 idealised simulations are identical except for the GIS elevation, which is decreased/increased by ±2% up to ±48% ($\beta$ scaling percentage - A1). This decrease/increase is applied evenly over the entire GIS. This simple method is used because it shows a well controlled and more comprehensible idealised framework for sensitivity studies about the dependence of $\delta^{18}O$ and climate on the magnitude and sign of GIS elevation changes.

To help isolate the impact of elevation changes, the present-day GIS extent is unmodified. Each elevation change simulation is time integrated for a total of 475-years which ensures appropriately spun-up atmosphere and upper ocean fields. In each case the final 50 years of the simulations are considered for averaging.

### 2.2.2 Joint impact of changes in sea ice and GIS morphology

A second ensemble of 32 simulations with different GIS configurations and sea ice retreat scenarios is used to help explore the joint impacts of sea ice change and GIS change on Greenland (See Fig. A1 and Fig. A2 and Table A1). This second set of simulations help us explore both changes in the extent (land-ice fraction) (Fig. A2) and elevation of the GIS (Fig. A1).

The methodology outlined in Domingo et al. (2020) is used to generate these simulations. The parameterisation of the set of GIS morphologies and sea ice retreat scenarios is performed by means of a Principal Component Analysis (PCA) approach. Due to the spherical geometry of the Earth, the application of a classical PCA to our data would be inappropriate. We thus apply a particular case of generalised PCA analysis (weighted PCA), described in Jolliffe (2002) and Salter et al. (2019). In summary, (1) GIS morphology modes are calculated from an initial ensemble of 14 LIG GIS reconstructions (Robinson et al., 2011; Born and Nisancioglu, 2012; Helsen et al., 2013; Quiquet et al., 2013; Stone et al., 2013; Calov et al., 2015; Langebroek and Nisancioglu, 2016), and (2) associated relevant sea ice retreat scenarios are calculated from 22 Arctic sea ice change experiments performed by Malmierca-Vallet et al. (2018) (See Fig. A1 and Table A1). In particular, we generate a set of 32 nine-dimensional random vectors. The first eight components of each vector are independently normally distributed and are used to generate a new GIS morphology via linear combination of the PCs (same procedure as in Domingo et al. (2020)). The ninth component represents instead heat flux, and is uniformly distributed between 0 and 120 $W/m^2$. We follow the methodology of Holloway et al. (2016a, 2017) and Malmierca-Vallet et al. (2018) on sea ice forcing and implement these heat fluxes to the bottom of the Arctic sea ice (See Table. A1). This sea ice forcing is kept constant through the complete annual cycle, and thus the model still calculates the seasonal cycle of sea ice growth and decay. Sea ice varies over time with the coupled model, and both the oceanic and atmospheric components of HadCM3 respond to variations in sea ice (for more details on the methodology, see Malmierca-Vallet et al. (2018)).

The resulting 32 LIG GIS morphologies show strong variation in terms of both height and ice extent (Fig. A1 and Fig. A2). Some morphologies show a rather small retreat of the GIS, and others a possible division of the GIS into two domes, some display strong ice loss in the south, while others show substantial ice retreat in the north.

After the design GIS morphologies and associated sea ice forcing are generated, HadCM3 is used to model the isotopic response to these modified GIS morphologies and sea ice retreat scenarios at the ice-core sites. All LIG experiments are forced with orbital parameters and GHG values appropriate for 125 ka, and time integrated for a total of 475-years.

## 3 Isotopic simulation results

We present results from (1) 16 GIS elevation change only scenarios and, (2) 32 experiments with combined Arctic sea ice forcing and modified GIS morphology (see Table A1). A two-sided Student's t-test is utilised to estimate the statistical significance

of changes (von Storch and Zwiers, 2001). Hereafter, lapse rates are defined to be positive, if the analysed atmospheric variable decreases with elevation.

## 3.1 GIS elevation change scenarios

### 3.1.1 Mean annual $\delta^{18}O$ changes at ice core sites

At the NEEM deposition site (around 205 ± 20 km upstream of the NEEM drill site due to ice flow; NEEM community members (2013)), the 125Control experiment simulates a precipitation-weighted $\delta^{18}O$ (hereafter $\delta^{18}O$p) anomaly of 1.9‰ compared to PI. The lowered GIS experiments have $\delta^{18}O$p anomalies which vary between 0.6‰ and 6.4‰ whilst the increased GIS elevation experiments act to decrease $\delta^{18}O$p anomalies by as much as -3.9‰ in the most extreme scenario p1300, relative to 125Control (Fig. 1a).

The 125Control experiment shows $\delta^{18}O$p anomalies of 1.5‰ at GRIP and GISP2 and 1.6‰ at NGRIP compared to PI. Depending on the prescribed reduction of the ice sheet elevation, $\delta^{18}O$p anomalies compared to 125Control vary between 0.5‰ and 6.5‰ at NGRIP (Fig. 1e), between 0.3‰ and 6.8‰ at GISP2 (Fig. 1m) and between 0.2‰ and 6.6‰ at GRIP (Fig. 1i). In contrast, the increased elevation scenarios show a decline in $\delta^{18}O$p anomalies of up to -3.6‰ at NGRIP and GRIP and -4.0‰ at GISP2 in the most extreme scenario p1300 relative to 125Control (Fig. 1e,i and m).

With respect to the PI simulation, the 125Control simulation shows a $\delta^{18}O$p rise of 1.0‰ at DYE3 and 1.8‰ at Camp Century. The decreased elevation scenarios present $\delta^{18}O$p anomalies ranging from 0.4‰ to 5.7‰ at DYE3 and from 0.7‰ to 4.6‰ at Camp Century compared to 125Control (Fig. 1q and u). $\delta^{18}O$p anomalies decrease up to -3.9‰ at DYE3 and -3.3‰ at Camp Century in p1300 compared to 125Control (Fig. 1q and u).

$\delta^{18}O$p anomalies are weaker for increases in GIS elevation than for decreases (Fig. 1). Core-averaged $\delta^{18}O$p anomalies are
20    -3.7‰ compared to 6.1‰ for a 1300m increase/decrease in relative elevation respectively. This results in a non-linear isotopic lapse rate across the elevation change scenarios (Fig. 1), which will be discussed in the following sections.

For the rest of the results section, for clarity, we focus particularly on two example scenarios which depict medium-high GIS elevation changes (experiments marked in red in Table A1).

### 3.1.2 Surface air temperatures

The orbital forcing dominates the climate in the 125Control simulation. In Greenland, summer local temperature increases exceed 3.5°C due to the large increase in summertime insolation (Fig. 2c).

The local surface climate over Greenland is noticeably affected by local changes in GIS surface elevation. Decreases in GIS elevation act to increase surface air temperatures (SATs) across Greenland, and vice versa (Fig. 1 and Fig. 2). In Greenland, the scenario with decreased elevation (m900) simulates positive SAT anomalies all year round compared with 125Control (Fig. 2d-f). Annual local temperature increases exceed 4.5°C in m900 relative to 125Control. As expected, the increased elevation scenario p900 shows negative SAT anomalies throughout the year relative to 125Control experiment (Fig. 2g-i). In central regions over Greenland, local temperature decreases exceed -4.5°C during both summer and winter seasons.

A previous study by Merz et al. (2014a) found that the sensitivity of LIG Greenland's climate to GIS topography changes is seasonally diverse. For example, in winter, strong cooling conditions are found over some areas that become ice-free and flat, while the remaining ice dome show warmer conditions (Merz et al., 2014a). Areas that become ice-free are characterised by weak surface winds and turbulence, barring an efficient sensible heat flux and leading to very strong temperature inversion (Merz et al., 2014a). Our idealised elevation change simulations do not show this temperature inversion mechanism; this is most likely linked to the unchanged land-ice distribution in our experiments.

Averaging across six ice core sites (Camp Century, NEEM, NGRIP, GRIP, GISP2 and DYE3), temperature lapse rates vary slightly from 0.47°C per 100 m for the lowered GIS states to 0.44°C per 100 m for the enlarged GIS states (Fig. 1 - blue solid lines), however these changes are not statistically significant.

### 3.1.3 Atmospheric circulation

To better understand the variations in atmospheric circulation that occur in response to changes in surface elevation we show changes in the low-level wind pattern (at 850 hPa) and mean sea level pressure (MSLP) field. The 125Control simulation exhibits a widespread decrease in summer (JJA) MSLP compared to PI (Fig. 3c); the warmer SATs and Arctic sea ice loss in 125Control result in a warmer and less stable atmosphere at northern high latitudes during summer. During both winter and summer, the 125Control experiment shows no major differences in wind direction or strength relative to PI (Fig. 4a-d and Fig. A3a-b).

Over the Norwegian Sea, there is an increase in winter MSLP (local increases exceed +50 Pa) in m900 relative to 125Control (Fig. 3e). This increase is coincident with a sea ice increase (Fig. 5d) and cooler SATs (Fig. 2e) over the same region compared to 125Control. Around northern Greenland, the scenario m900 shows a decrease in summer MSLP relative to 125Control (local decreases exceed -50 Pa; Fig. 3f); this is coincident with an decline in sea ice concentration (Fig. 5a). The scenario p900 shows and increase in annual MSLP over central Arctic Ocean (Fig. 3g).

Over Greenland, the surface winds respond to variations in the GIS surface elevation. Strong anticyclonic flow centred over Greenland is characteristic of the PI and 125Control simulations (Fig. 4 and Fig. A3). In the decreased elevation experiment (m900), the Greenland anticyclone becomes smaller (especially during the winter months compared to 125Control; Fig. 4c-f and Fig. A3c) and, over northeast Greenland, local wind vectors suggest air masses inflow from the Arctic Ocean contrary to the common outflow observed in the PI and 125Control (Fig. 4 and Fig. A3c). In contrast, the scenario with increases in the GIS elevation (p900) display an enhanced anticyclonic flow particularly during winter (Fig. 4g-h and Fig. A3e).

### 3.1.4 Changes in precipitation pattern

During summer, the 125Control shows an enhanced precipitation rate compared to PI mainly across southwestern and central Greenland (Fig. 6c). This is in line with results from other climate models (e.g., Otto-Bliesner et al., 2006; Merz et al., 2014b).

There is a rise in precipitation rate over much of Greenland throughout the year in m900 compared to 125Control (Fig. 6d to f). This is expected as the lowering of the orography leads to a wider spread of precipitation across Greenland from the east and west which is blocked by the higher and steeper elevation of the present-day GIS. Local increases over south-east Greenland

exceed 0.8 mm/day in m900 during winter (Fig. 6e). This increase in precipitation accords with a reduction in winter sea ice concentration along the east coast of Greenland relative to 125Control (Fig. 5d).

For the increased elevation scenarios, local changes in precipitation rate relative to 125Control are less widespread and smaller than for the decreased elevation scenarios during both seasons (Fig. 6 d-i). Over south-east Greenland, p900 is up to
0.6 mm/day drier than the 125Control simulation during winter (Fig. 6 h).

Precipitation increases linked to elevation decreases are much larger than the drying linked to elevation increases, implying non-linearities in the climate response to GIS elevation change (Fig. 1 and Fig. 6). The core-average (averaging across the six core sites: NEEM, NGRIP, GRIP, GISP2, DYE3 and Camp Century) precipitation lapse rate varies from 0.097 mm/year per 100 m for the decreased GIS elevation states to 0.009 mm/year per 100 m for the increased GIS elevation states (Fig. 1 - blue
solid lines). Nevertheless, this is considerably influenced by the DYE3 and Camp Century ice core sites (Fig. 1 s and w - blue solid lines). The DYE3 ice core site shows a much steeper relationship than the other ice core sites, 0.26 mm/year per 100 m for the decreased elevation scenarios and 0.134 mm/year per 100 m for the increased elevation scenarios (Fig. 1 s - blue solid lines). At Camp Century site, precipitation tends to increase with increasing surface elevation (-0.091 mm/year per 100 m elevation increase) (Fig. 1 w - blue solid line). Removing the Camp Century site increases the core-average precipitation
gradient to 0.029 mm/year per 100 m elevation increase. The different behaviour found at Camp Century site is likely linked to the orographic enhancement of precipitation (Johnson and Hanson, 1995; Frei and Schär, 1998; Petersen et al., 2004; Roe and Baker, 2006) among the enlarged GIS states. Reductions in Camp Century height results also in marginal increases in precipitation rate (0.05 mm/year per 100 m) which are probably related to the weakening of the Greenland anticyclone and smaller barrier effect.

### 3.1.5   Changes in sea ice

For the PI simulation, the September mean sea ice extent is $5.8 \times 10^6 km^2$. The 125Control simulation shows a reduced September mean of $4.4 \times 10^6 km^2$ relative to PI; larger seasonal and latitudinal insolation variations (linked to the orbital forcing) lead to Arctic sea ice loss during summer/spring (e.g., Otto-Bliesner et al., 2006).

GIS elevation reductions lead to an increase in winter sea ice extent, whereas increases in the GIS elevation result in winter
sea ice retreat (Fig. 1). Thus, the loss/increase of winter sea ice extent act as a positive/negative feedback on $\delta^{18}O$. In contrast to $\delta^{18}O$ and SAT, variations in winter sea ice extent are smaller for decreases in GIS elevation compared to increase elevation scenarios. For example, the March sea ice extent is reduced by $-4.2\%$ in p900 and increased by $+1.7\%$ in m900 compared to the 125Control simulation.

The decreased elevation scenario (m900) displays an increase of winter sea ice concentration on the Norwegian Seas and
on the southern-eastern coast of Greenland compared to 125Control simulation (Fig. 5 d). The reduced cyclogenesis off the south-east coast of Greenland (Fig. A5), results in growth of winter sea ice over these regions (Fig. 5 d). This is probably associated with a decrease in wind-driven ocean heat transport (e.g., Pausata et al., 2011; Stone and Lunt, 2013; Davini et al., 2015). The increased elevation scenario (p900) experience the same forcing but in opposite direction (Fig. 5 f and Fig. A5).

We also find some local changes in summer sea ice concentration; while p900 shows decreases of summer sea ice over the Beaufort Sea, it shows increases over the Fram Strait area. Similar patterns are found in m900 but in opposite direction and of lower magnitude (Fig. 5 a,c).

We ascribe these changes in summer sea ice to variations in ocean salinity caused by anomalous downwelling or upwelling, induced by anomalously low or high sea level pressure over the Arctic (Jackson and Vellinga, 2012). In HadCM3, the geostrophic balance of the Beaufort gyre can be altered ageostrophically by wind stresses linked to low-frequency sea level pressure variability (Jackson and Vellinga, 2012). Our increased elevation scenario (p900) show high sea level pressure anomalies over the Arctic basin (Fig. 3) which lead to downwelling in the center of the Arctic basin and upwelling along the coasts respectively (Fig. A4). Since the surface water is fresher and colder than the subsurface water, this results in salinification near the coasts and freshening in the center of basin (Fig. A4). The same mechanisms apply to the decreased elevation scenario (m900) but in opposite direction (Fig. A4).

The increase in wind speed along the Fram Strait in p900 compared to 125Control and vice versa for m900 (Fig. A5) also affects the advection of sea ice from the Arctic to the Atlantic ocean (Davini et al. (2015)).

## 3.2   The response of the isotopic lapse rate to changes in the background climate state

Malmierca-Vallet et al. (2018) demonstrate the importance of Arctic sea ice changes as a control on LIG Greenland ice core $\delta^{18}O$ because of its impact on both the regional temperature increase and the moisture source. Thus, we also study 32 simulations that examine the joint impact of modified Arctic sea ice retreat and modified GIS morphology (considering both changes in the extent and elevation of the GIS - see Table A1 and section 2.1).

We use the sea ice retreat simulations of Malmierca-Vallet et al. (2018) to isolate the impacts of $\delta^{18}O$ due to sea ice variation. This allows to test to which extent Arctic sea ice changes may influence isotopic lapse rate values. Fig A6 shows the change in $\delta^{18}O$ as a function of winter (March) sea ice retreat. We (1) remove the orbital forcing effect by calculating $\delta^{18}O$ anomalies compared to the 125 ka control simulation, and (2) we only analyse scenarios with winter sea ice retreats lower than $55\%$, due to the almost no sensitivity of Greenland $\delta^{18}O$ to sea ice losses greater than $50\%$ (for more detail see Malmierca-Vallet et al. (2018)).

To calculate the sea-ice-corrected $\delta^{18}O$ anomalies we deduct the sea-ice-associated $\delta^{18}O$ effect from the total $\delta^{18}O$ anomalies. Fig 7 shows the resulting sea-ice-corrected $\delta^{18}O$ anomalies (Fig 7 – purple curve fits) as well as total $\delta^{18}Op$ anomalies (compared to both the PI and 125Control simulations) not corrected for sea ice changes (Fig 7 – red and blue curve fits respectively) for both sets of simulations (Fig 7 - triangles for elevation change simulations and dots for simulations looking at the combined impact of sea ice retreat and GIS changes).

When considering total $\delta^{18}Op$ anomalies (relative to 125Control) not corrected for sea ice changes, a non-linear $\delta^{18}O$ lapse rate is observed over Greenland (Fig 7 - second column); The core-average $\delta^{18}O$ lapse rate varies from $0.29\%_o$ per 100 m for the enlarged GIS states to $0.49\%_o$ per 100 m for the lowered GIS states (Fig 7 – second column). These results thus strongly suggest a non-linearity in the isotope-elevation relationship, with higher $\delta^{18}O$-elevation gradients for lowered GIS states and vice versa.

When further deducting the winter sea ice effect, we find an almost stationary core-average $\delta^{18}O$ lapse rate, slightly varying from 0.38‰ per 100m for the enlarged GIS scenarios to 0.39‰ per 100m for the lowered GIS scenarios (Fig 7 - third column). The sea ice effect increases $\delta^{18}O$-elevation gradients by $28\%$ in the enlarged GIS states and decreases $\delta^{18}O$-elevation gradients by $-19\%$ in the lowered GIS states. Indeed, this suggests that sea ice changes may strongly influence linearity in the isotope-
elevation relationship.

## 4   Discussion

### 4.1   Response of Arctic sea ice and atmospheric circulation to GIS elevation changes

Our lowered GIS experiments show similar climate behaviour to previous studies where GIS is removed (Toniazzo et al., 2004; Stone and Lunt, 2013; Davini et al., 2015). During summer warming over the GIS is enhanced due to its lower elevation and
Arctic sea ice retreat (Lunt et al., 2004). During winter warm anomalies extend over the Arctic Ocean. These anomalies are related to a smaller anticyclone over Greenland (Fig. 4h) (Stone and Lunt, 2013; Davini et al., 2015). Support comes also from Merz et al. (2014a) who show, for their perturbed LIG experiments with reduced GIS, a smaller anticyclone as well as decreased wind velocities. The lowering of the elevation also leads to a weakened ice sheet barrier effect, permitting cyclonic systems to get into more central and northern areas of Greenland. These modelled results are in agreement with the findings of
Merz et al. (2014a) and Hakuba et al. (2012) who show that a decrease in the height and size of the GIS weakens the barrier effect, permitting more moisture to be advected to the plateau. The reduction in cyclogenesis over the Norwegian Sea and off the south-east coast of Greenland due to lowering of the GIS elevation leads to the growth of further sea ice, especially during the winter months (Fig. 5g and h). Reduced ocean heat transport, due to weakened wind-driven currents (Stone and Lunt, 2013) may also contribute to decreased surface temperatures over the Norwegian and Barents Seas and on the south-east coast
of Greenland (Fig. 2 e and h).

When elevation is increased, colder GIS temperatures occur, but a compensating warming occurs around Greenland (Fig. 2). Similar mechanisms causing this surface temperature pattern were discussed in Merz et al. (2014a) who investigated the sensitivity of LIG Greenland's climate to GIS topography changes. In particular, Merz et al. (2014a) found cooling over areas of higher elevation (eastern Greenland) but warming on the periphery of the ice sheet. GIS topography changes influence the
Greenland's surface energy balance through changes in surface winds and turbulent heat fluxes (Merz et al., 2014a). In our experiments, the glacial anticyclone over Greenland intensify as elevation increases, especially during the winter months (Fig. 4i). This leads to enhanced cyclogenesis over the Barents Seas and south-east coast of Greenland which result in winter sea ice retreat over the same regions (Fig. 5f).

## 4.2 Lapse rates in response to GIS elevation changes

Averaging across the six ice core sites, we find temperature lapse rates of 0.47°C and 0.44°C per 100m for the decreased/increased elevation change scenarios respectively (Fig. 1). We also find that the wetting related to decreases in GIS elevation are higher than the drying related to increases in elevation.

Our temperature lapse rates compare well with previous estimates which show that the near-surface temperature lapse rate can generally differ from the free-air lapse rate (gradient of moist adiabatic cooling of 0.65°C per 100m) (Marshall et al., 2007; Fausto et al., 2009; Gardner et al., 2009; Erokhina et al., 2017). Marshall et al. (2007) monitored 25 sites with altitudes between 130 to 2010 m across the Prince of Wales Icefield and observed a mean daily temperature lapse rate of 0.41°C per 100 m. Furthermore, Gardner et al. (2009) showed temperature lapse rates near four glacier surfaces, in the Canadian high

Arctic, of 0.49°C per 100 m (ablation season mean). Gardner et al. (2009) also suggested that lower temperature lapse rates are expected under a warming climate and linked this to the negative relationship found between lower-troposphere temperatures and slope lapse rates.

## 4.3 The response of the isotopic lapse rate to the background climate state

Isotope-elevation gradients have tended to be calculated from modern surface data (e.g., Dansgaard, 1973): a present-day spatial

relationship is presumed to apply to temporal changes. This disregards any impact that variations in the ice sheet elevation may have on the atmospheric circulation, precipitation patterns and eventually the isotopic composition.

     Our idealized elevation change simulations with HadCM3 allow a fuller calculation. We find a smaller core-average $\delta^{18}O$ lapse rate for enlarged GIS states (0.29‰ per 100 m) than for the lowered GIS states (0.49‰ per 100 m) (Fig 7). Hence, $\delta^{18}O$-elevation gradients do not remain constant across the parameter space of elevation changes. This strongly suggest non-

linearities in the isotopic response to Greenland elevation change.

     We also find that winter sea ice variations can increase/decrease modelled core-averaged isotopic lapse rate values by about $+28\%$ and $-19\%$ for the enlarged/lowered GIS states respectively (Fig. 7). These results thus suggest that sea ice variations may have a strong influence on $\delta^{18}O$-elevation gradients, especially at coastal areas such as the Camp century ice core site (Fig. 9). In particular, at this location, we find that the winter sea ice effect decreases the $\delta^{18}O$-elevation gradient by $-24\%$

in the lowered GIS states and increases the $\delta^{18}O$-elevation gradient by as much as $92\%$ in the enlarged GIS states (Fig. 9 e). While the largest influence of sea ice on $\delta^{18}O$ changes is found at Camp Century site, DYE3 site shows the smallest (Fig. 9 e-f). These results point to elevation changes as a likely driver (together with GHGs and orbital forcing) on LIG $\delta^{18}O$ changes at DYE3 ice core site. This is in agreement with previous LIG GIS modelling studies which propose a significant LIG lowering around the DYE3 area, even the total loss of ice (Robinson et al., 2011; Helsen et al., 2013).

Interestingly, LIG isotopic lapse rates and the PI spatially derived isotopic lapse (0.37‰ per 100 m) modelled with HadCM3, are lower than the modern spatially derived gradients of 0.62‰ per 100 m and 0.72‰ per 100 m in central and northwest Greenland respectively (Dansgaard, 1973; Johnsen et al., 1989; Vinther et al., 2009). Furthermore, our LIG core-average $\delta^{18}O$ lapse rates are also somewhat lower than the lapse rate of 0.56‰ per 100 m modelled (with the isotope enabled version of

the European Centre Hamburg Model version 4) over Greenland for the LIG period by Sjolte et al. (2014). Note our modelled isotopic lapse rates contemplate the dynamical response of atmospheric circulation to GIS elevation changes and Arctic sea ice variations, whereas previous studies disregard these effects.

These elevation change simulation results thus suggest possible non-linearity in isotope-elevation gradients. It would be useful if this was checked with other models to assess model-dependence in the results. The HadCM3 resolution does not permit it to represent the steep GIS margins; this may be behind some of the model-data mismatches (Toniazzo et al., 2004).

## 4.4 Implications for NEEM $\delta^{18}O$ and elevation reconstructions

Considering the NEEM elevation reconstruction, which indicates NEEM elevation differences of +45±350 m at 126 ka relative to present-day (NEEM community members, 2013), we find a most likely increase in $\delta^{18}O$ values of between +0.8‰ to +3.5‰ relative to PI (Fig. 8). This relatively falls within the lower end of the uncertainty range of the reconstruction by Domingo et al. (2020): most likely LIG $\delta^{18}O$ peak of +3.6‰ and uncertainty range between +2.7‰ to +4‰. The relatively small overlap between the $\delta^{18}O$ record and the elevation reconstruction has already been discussed in Domingo et al. (2020) and, could possibly reflect uncertainties attached to the air content NEEM elevation reconstruction method. The methodology depends on making corrections to air content measurements related to insolation and temperature in conjunction with secular variations in surface pressure and winds (Martinerie et al., 1994; Krinner et al., 2000; Raynaud et al., 2007; Eicher et al., 2016). In addition, NEEM air content measurements between 127 and 118.3 ka are known to be affected by surface melting (NEEM community members, 2013).

## 4.5 Relative influence of sea ice and GIS changes on ice core $\delta^{18}O$

Considering the maximum reduction in NEEM's surface elevation proposed by the NEEM community members (2013) of -305 m at 126 ka, we find that the impact of LIG orbital-sea ice changes appears to be the dominant factor determining $\delta^{18}O$ changes (explaining 60% of the $\delta^{18}O$ anomaly), followed by GIS-driven sea ice changes (Fig. 9a). This is in agreement with previous studies that show the importance of changes in GIS topography and sea ice retreat to explain the LIG warming at the NEEM ice core site (Merz et al., 2014a, 2016; Guarino et al., 2020).

To make a comparable analysis at the other ice core sites, of the relative influence of each factor on determining $\delta^{18}O$ changes, we consider the same reduction in surface elevation of -305 m at the other locations (Fig. 9). We find that LIG orbital-sea ice changes is the dominant factor determining $\delta^{18}O$ changes at NGRIP, GRIP and GISP2 (accounting for 55-58% of the $\delta^{18}O$ changes) (Fig. 9b-d), while ice sheet changes appears to have the largest impact on $\delta^{18}O$ changes at DYE3 site (accounting for 48% of the $\delta^{18}O$ anomaly) (Fig. 9f). The highest sea ice influence is found at Camp Century (explain 10% of the $\delta^{18}O$ changes) (Fig. 9e).

Note, the above-mentioned relative influence of each parameter on $\delta^{18}O$ changes should be interpreted with caution; these results could change substantially if we were to consider any other possible elevation change scenario. There is no independent gas content information on elevation changes for DYE3 and Camp Century. Moreover, although there is total air content

records that were measured on the GRIP (Raynaud et al., 1997) and NGRIP (Eicher et al., 2016) ice cores, the authors show how complex it is to interpret this proxy in term of elevation changes at the drilling site.

Additional data on elevation changes together with better dated ice, especially at DYE3 and Camp Century sites, would be particularly valuable to further assess our quantitative elevation change scenarios. In addition, considering sea ice and ice changes in a joint framework following a Gaussian Process emulation approach (Domingo et al., 2020) and take also account of isostatic change may permit a valuable quantitative assessment of how changes in the GIS affected LIG global sea levels.

## 5 Conclusions

The results of this study are relevant for the interpretation of past climates from Greenland ice core records. Changing GIS elevation in HadCM3 alters the NH atmospheric circulation circulation, sea ice and precipitation patterns over Greenland and further afield. These climate feedbacks result in lower isotopic-elevation gradients for enlarged GIS states, and vice versa. Our results thus point to non-linear $\delta^{18}O$-elevation gradients over Greenland. We further show that isotopic lapse rate values may be significantly influenced by the background climate, in particular, winter sea ice changes.

These model results highlight the importance of the dynamical response of atmospheric circulation to GIS elevation changes when using isotopic measurements to derive past elevation changes: there may be non-linearities in isotope-elevation relationships. Although the underlying mechanism need further investigation, our finding has important implications for paleoclimate studies, in which stationary lapse rates are assumed and are normally based on present-day observations. Inter-model comparison studies would be helpful in further developing our understating of the isotope-elevation gradient over Greenland and how it varies with the background climate state.

*Code and data availability.* Access to the Met Office Unified Model source code is available under licence from the Met Office at https://www.metoffice.gov.uk/research/collaboration/um-partnership. The climate model data are available on request from http://www.bridge.bris.ac.uk/resources/simulations.

*Competing interests.* No competing interests are present

*Acknowledgements.* IMV acknowledges a NERC GW4+ studentship. The project has received funding from the European Union's Horizon 2020 research and innovation programme under grant agreement No 820970. L.C.S. acknowledges support through NE/P013279/1 and NE/P009271/1. This work used the ARCHER UK National Supercomputing Service (http://www.archer.ac.uk) and the JASMIN data analysis platform (http://jasmin.ac.uk/)

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

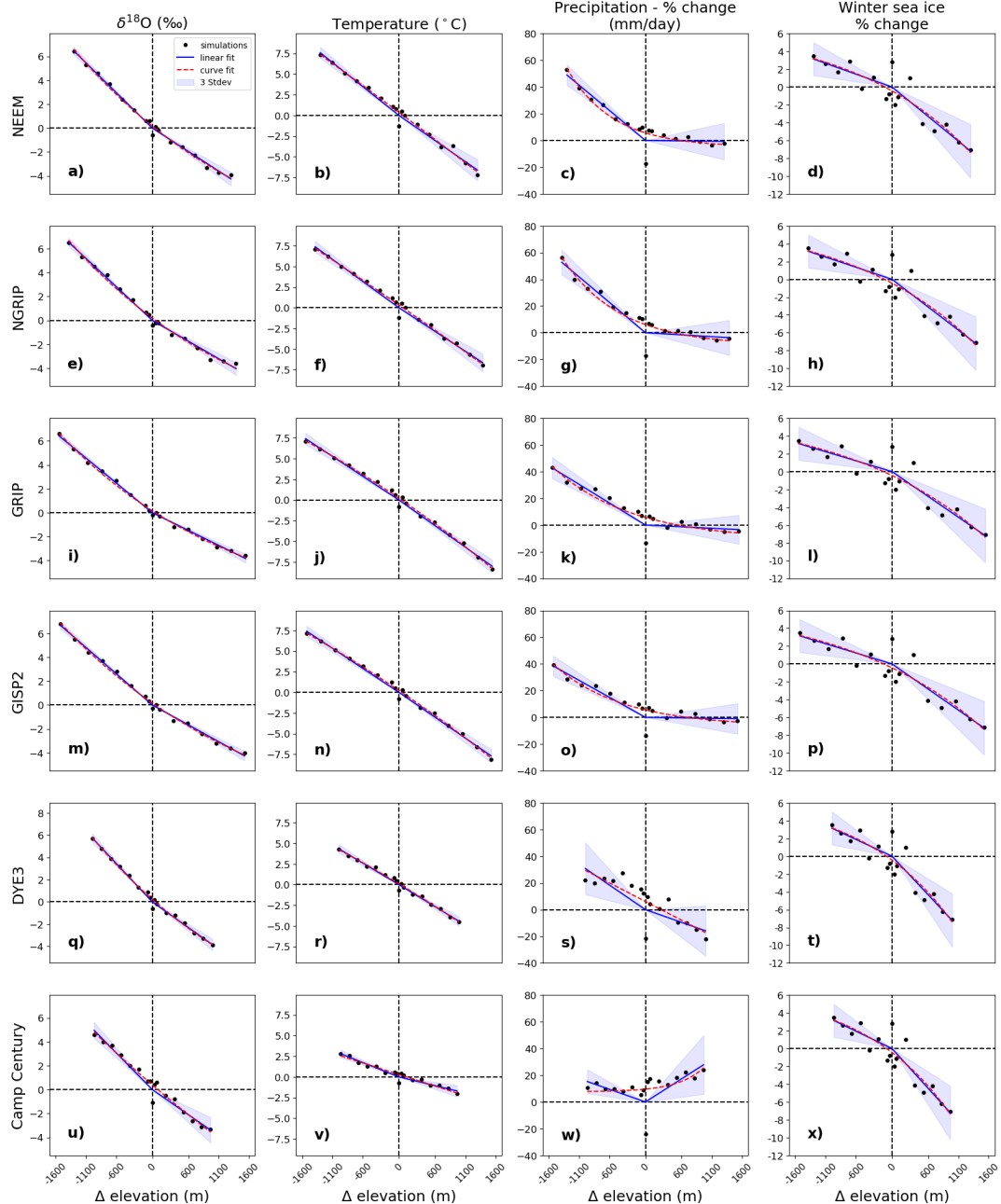

**Figure 1.** Change in $\delta^{18}O$p, temperature, precipitation flux and winter (March) sea ice extent as a function of GIS elevation changes (m). Changes are calculated as anomalies compared to 125 ka control experiment. Ice core sites displayed: (a-d) NEEM, (e-h) NGRIP, (i-l) GRIP, (m-p) GISP2, (q-t) DYE3, (u-x) Camp Century. Results for each of the 16 GIS elevation change scenarios are represented by black dots. Blue solid lines signify best fit lines (y = ax). Also shown ±3 Stdev (blue shade envelopes) on the best fit lines. Additionally, a second fit of the data using an exponential function (y = $\pm ae^{-bx} + c$) is included (red lines with dashes). Note: In the last column of plots, winter sea ice extent vs GIS elevation changes differs for the various ice core sites because of the different elevation changes at each ice core site compared to the 125 ka control.

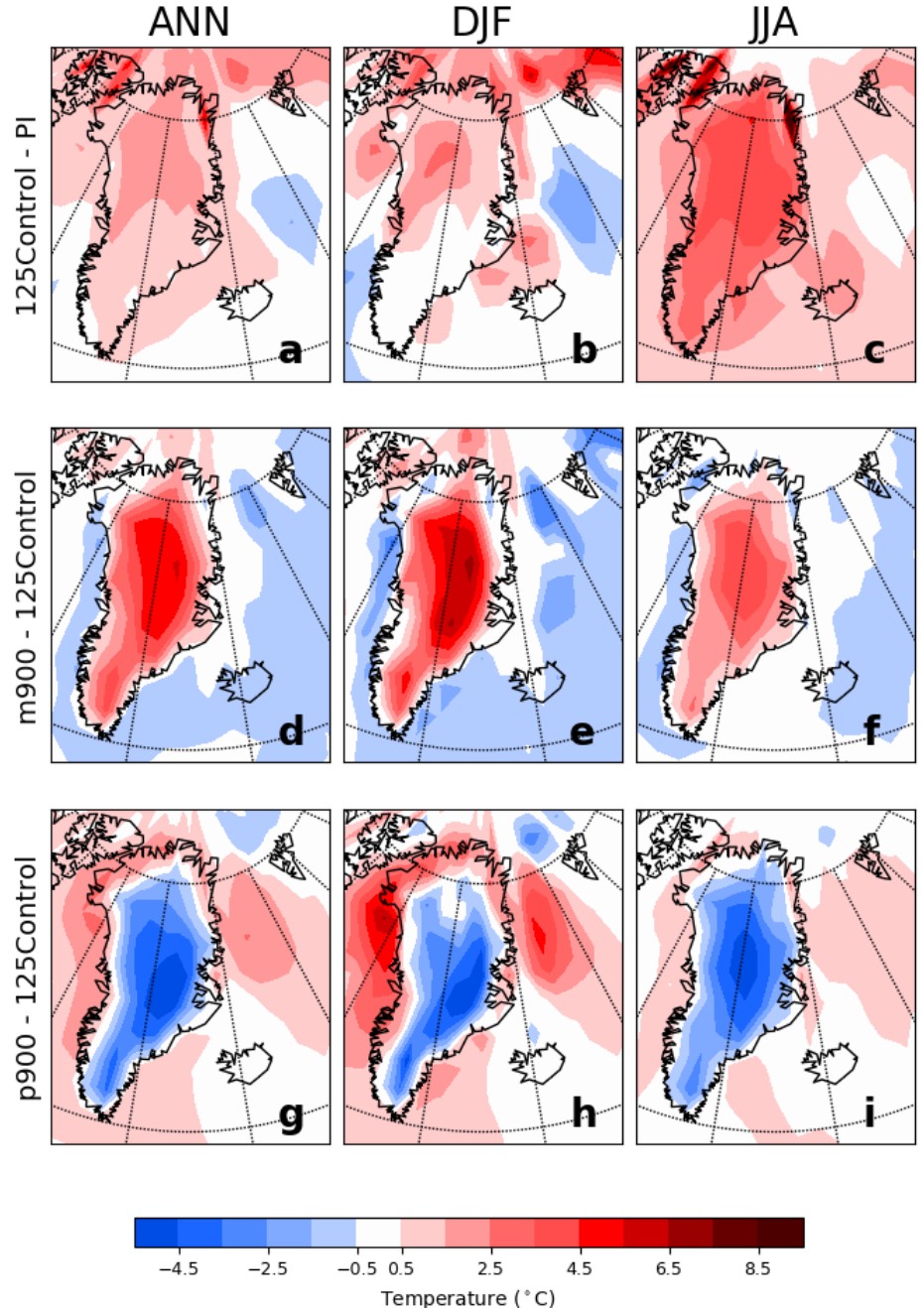

**Figure 2.** Modelled annual (ANN), winter (DJF) and summer (JJA) surface air temperature anomalies for m900 (d to f), and p900 (g to i) compared to the 125Control simulation. Also shown temperature anomalies for the 125Control compared to the PI simulation (a to c). Only the anomalies statistically significant at the 95% confidence level are displayed.

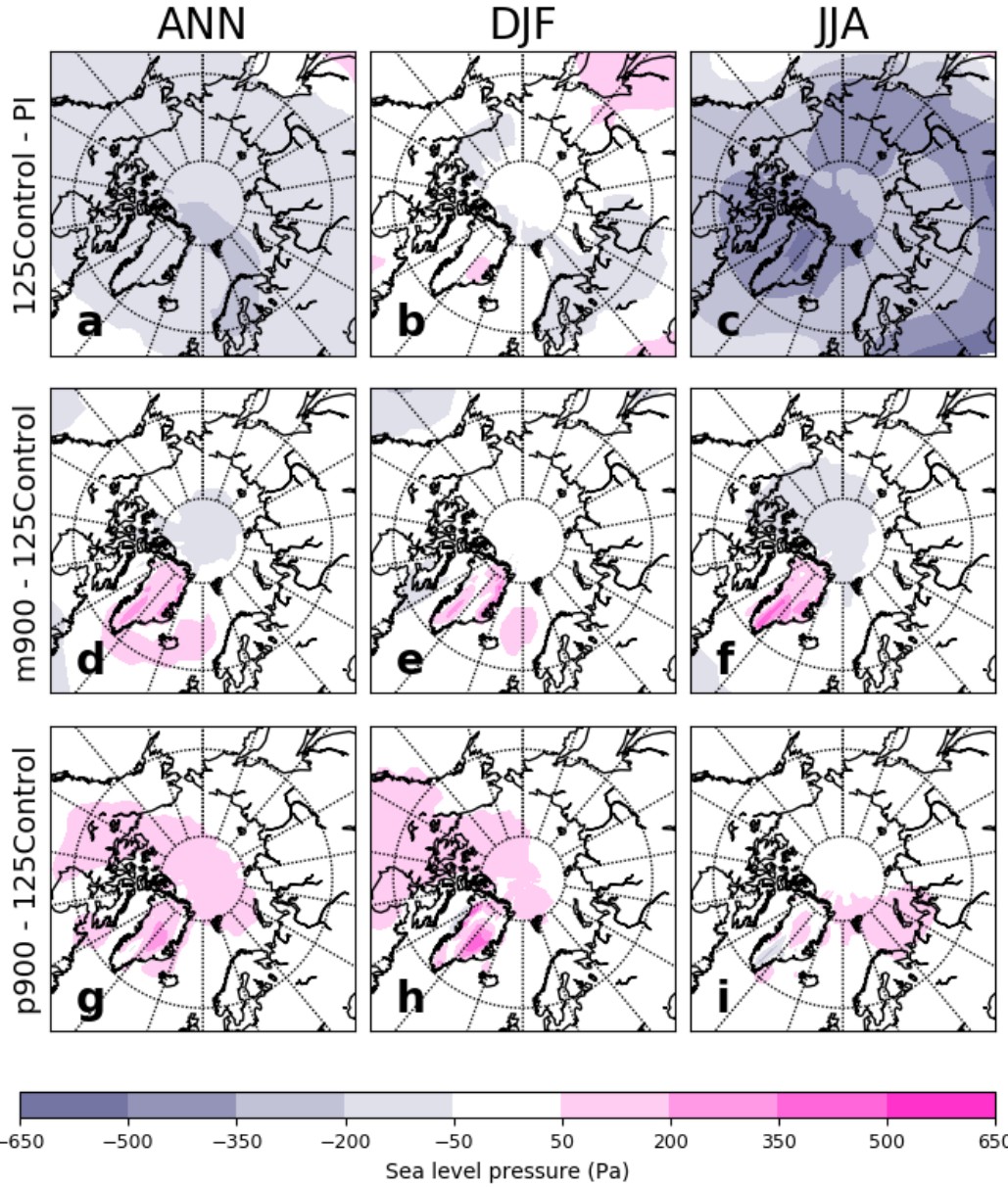

**Figure 3.** ANN, DJF and JJA mean sea level pressure anomalies (Pa) for: m900 (d to f) and p900 (g to i) compared to the 125Control simulation. Also shown sea level pressure anomalies for the 125Control compared to the PI simulation (a to c). Only the anomalies statistically significant at the 95% confidence level are displayed.

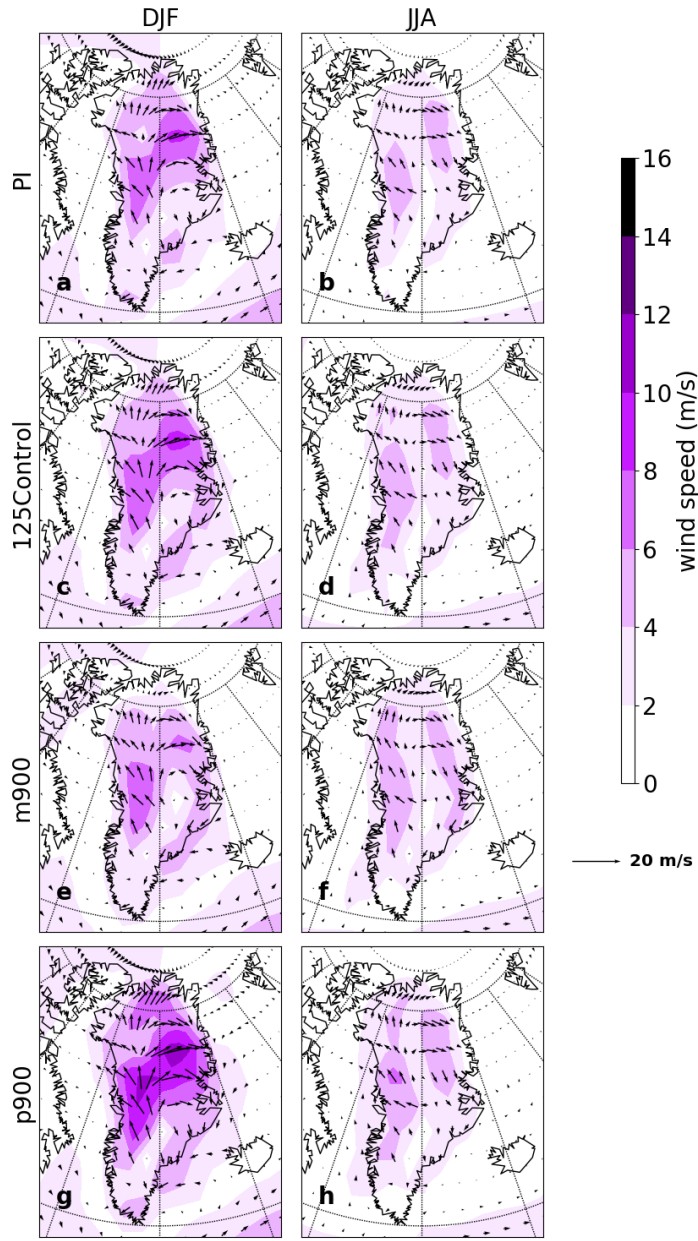

**Figure 4.** Absolute DJF and JJA low-level winds (at 850hPa) for PI (a-b), 125Control (c-d), m900 (e-f) and p900 (g-h). Shading displays wind speed (m/s).

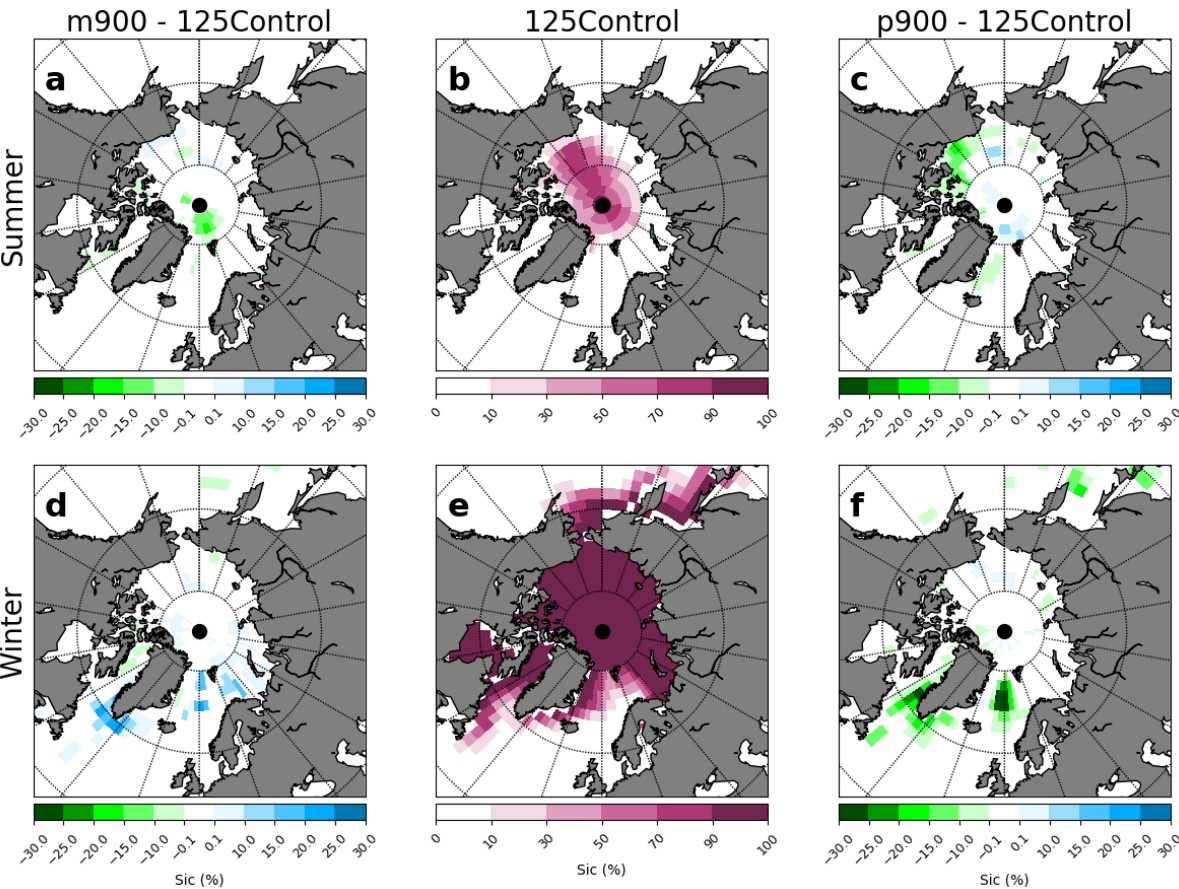

**Figure 5.** Sea ice concentration anomalies (%) for summer (September) and winter (March) for the scenarios m900 (a and d) and p900 (c and f) compared to the 125Control simulation. Also shown absolute sea ice concentration for the 125Control simulation for summer (b) and winter (e). Only the anomalies statistically significant at the 95% confidence level are displayed.

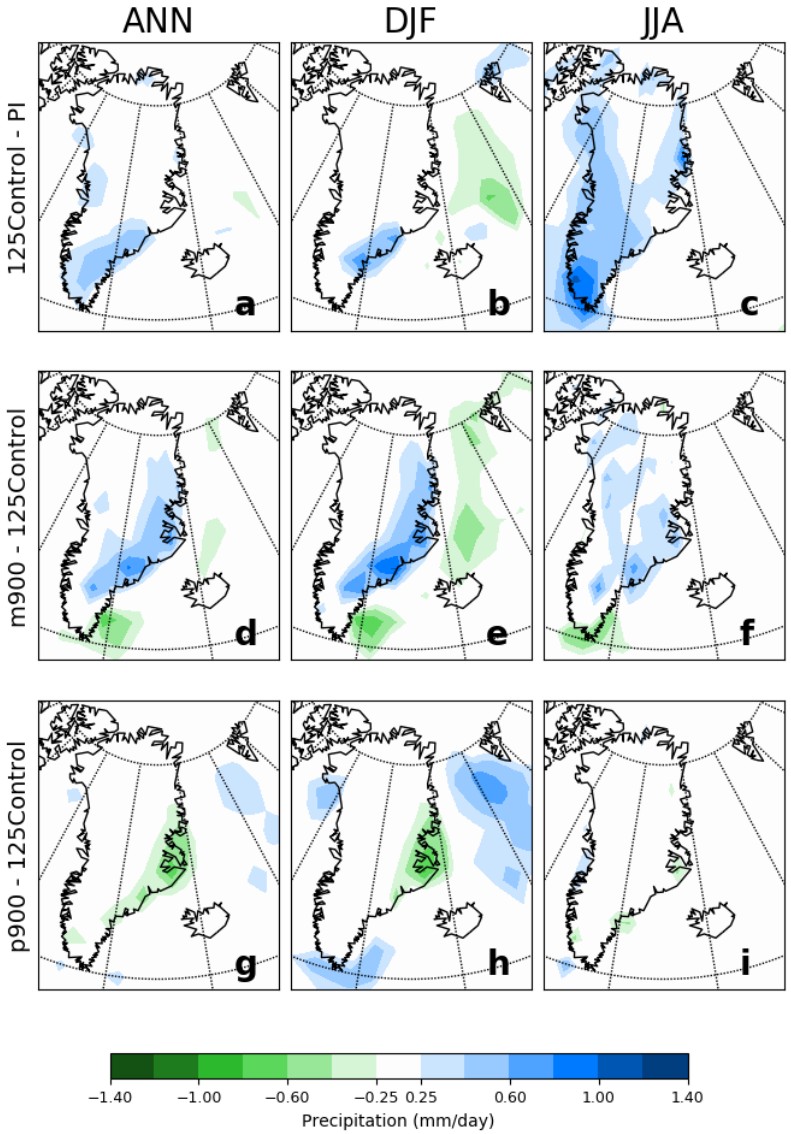

**Figure 6.** Annual (ANN), Winter (DJF) and summer (JJA) precipitation anomalies for m900 (d to f) and p900 (g to i) compared to the 125Control simulation. Also shown precipitation anomalies for the 125Control compared to the PI simulation (a to c). Only the anomalies statistically significant at the 95% confidence level are displayed.

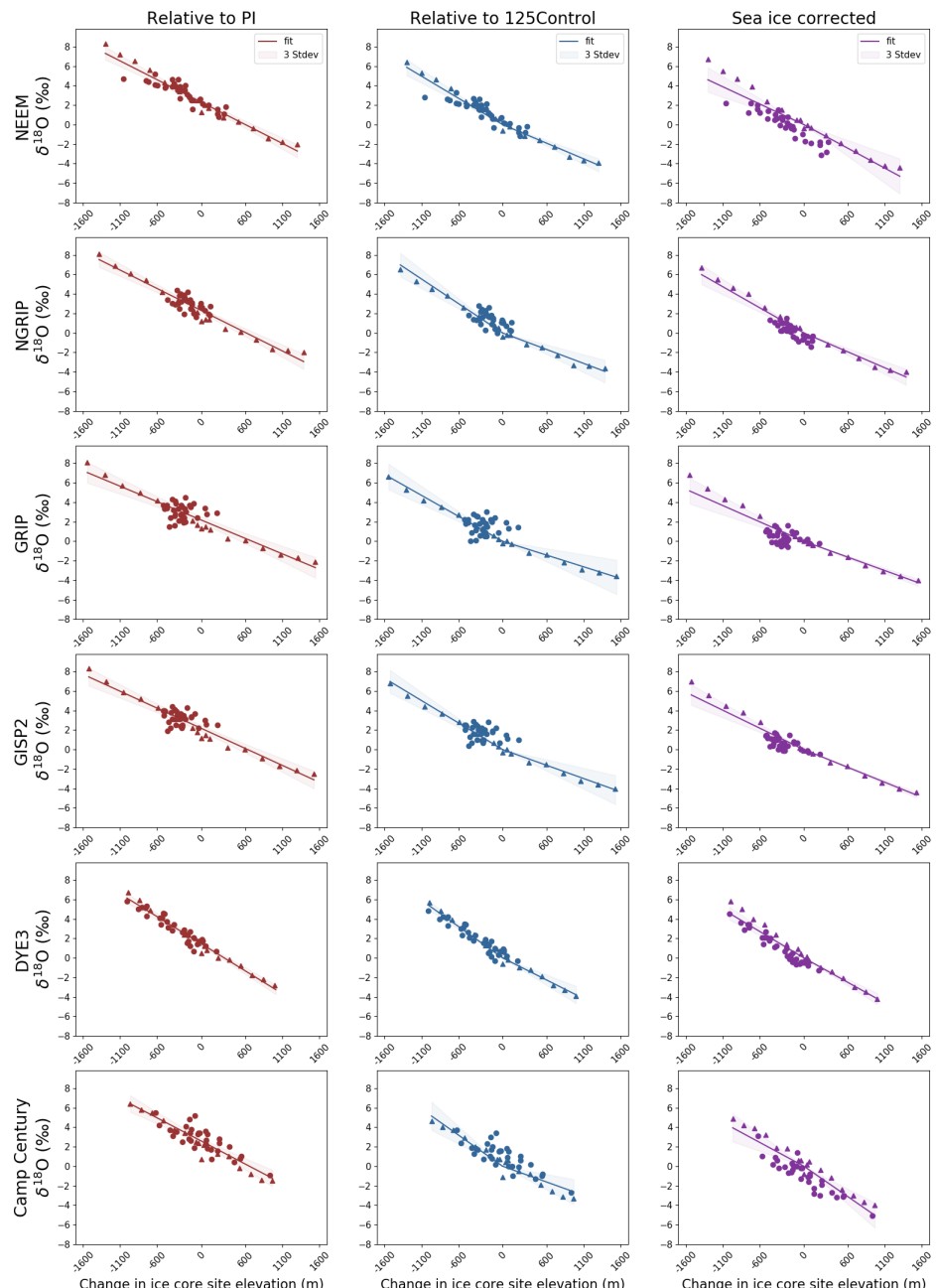

**Figure 7.** $\delta^{18}O$p anomalies as a function of ice core site elevation change (m) relative to PI (first column) and 125Control (second column). Also shown sea-ice-corrected $\delta^{18}O$p anomalies compared to 125Control (third row). Ice core sites displayed: NEEM (first row), NGRIP (second row), GRIP (third row), GISP2 (fourth row), DYE3 (fifth row) and Camp Century (sixth row). Triangles represent results for the 16 elevation change experiments. Dots represent results for the 32 simulations that examine the joined impact of Arctic sea ice retreat and modified GIS shape. Solid lines signify best fit curves (first column, y = a + bx; second and third columns, y = ax) and shade envelopes represent ±3s uncertainty on the best fit lines.

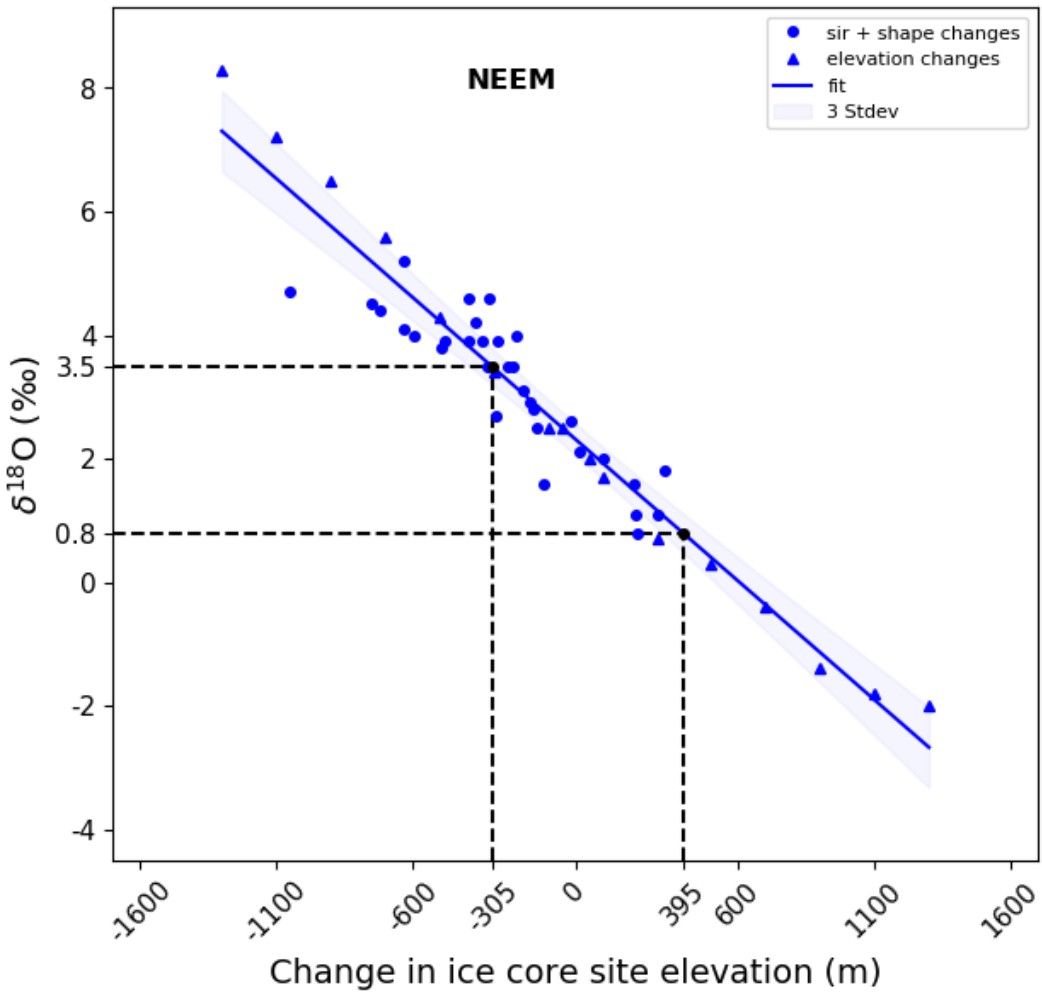

**Figure 8.** $\delta^{18}O$p anomalies as a function of the NEEM ice core site elevation change (m) relative to PI. Solid lines signify best fit curves (y = a + bx) and shade envelopes represent ±3 Stdev on the best fit lines. Triangles represent results for the 16 elevation change experiments. Dots represent results for the 32 simulations that examine the joined impact of Arctic sea ice retreat and modified GIS shape.

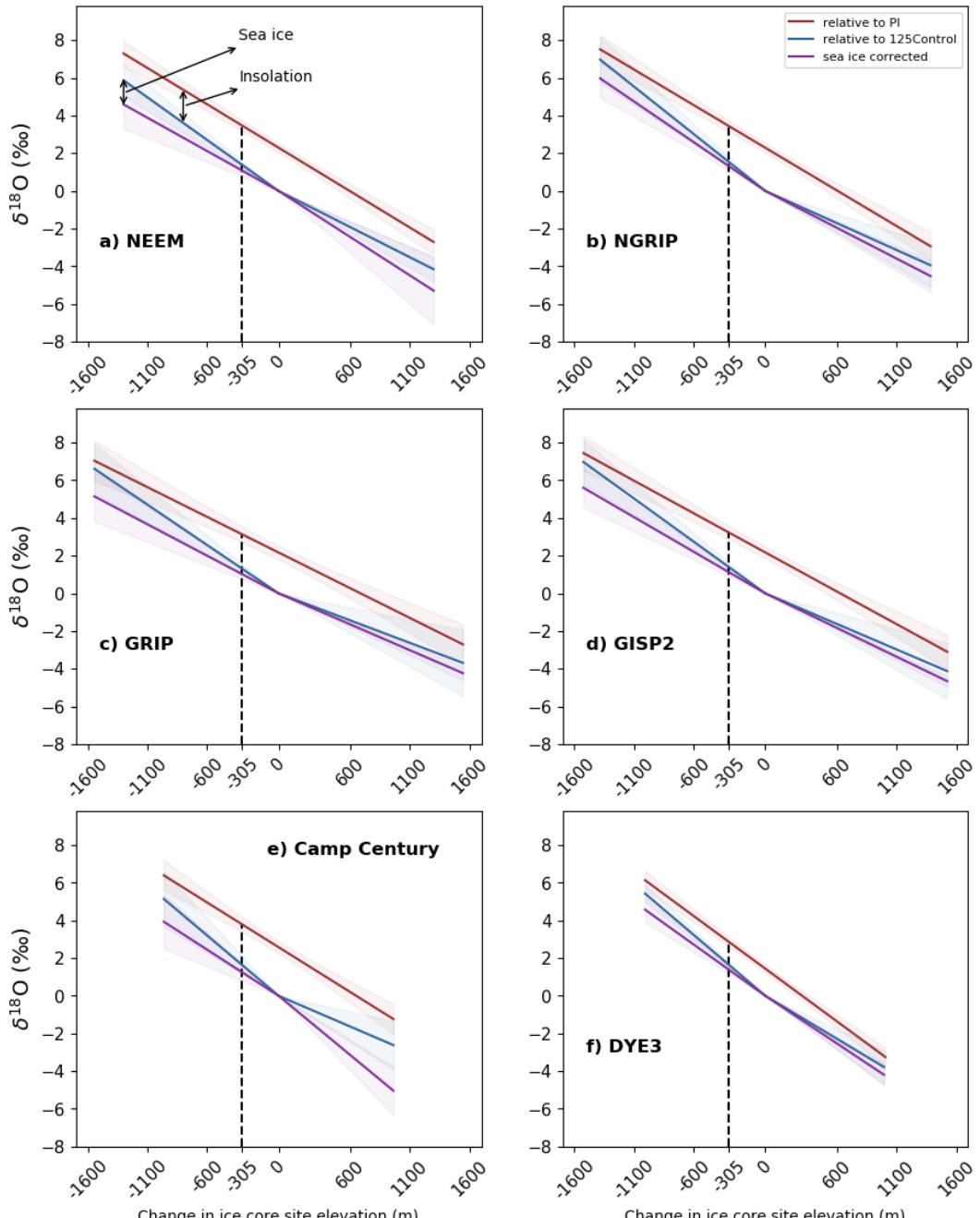

**Figure 9.** $\delta^{18}O$p anomalies as a function of the ice core site elevation change (m) relative to (1) PI (red), and (2) 125Control (blue). Also shown sea-ice-corrected $\delta^{18}O$p anomalies relative to 125Control (purple). Ice core sites shown: (a) NEEM, (b) NGRIP, (c) GRIP, (d) GISP2, (e) Camp Century, (f) DYE3. Solid lines signify best fit curves and shade envelopes represent ±3 Stdev on the best fit lines. Arrows indicate the influence of insolation and sea ice changes on $\delta^{18}O$ changes. In addition, the purple line (sea-ice-corrected $\delta^{18}O$p anomalies) represent the influence of GIS changes and associated atmospheric circulation changes on $\delta^{18}O$ changes.

**Appendix A**

Table A1: Full list of scenarios. The simulations highlighted in red are the ones primarily discussed in the text. All LIG simulations are performed with greenhouse-gas and orbital forcing centred at 125ka (Eccentricity - 0.04001; Obliquity - 23.80°; Perihelion - 201.3 day of yr; $CO_2$ - 276 ppmv; $CH_4$ - 640 ppbv; $N_2O$ - 263 ppbv). The boundary conditions for the PI simulation are the following: Eccentricity - 0.0167; Obliquity - 23.45°; Perihelion - 1.7 day of yr; $CO_2$ - 280 ppmv; $CH_4$ - 760 ppbv; $N_2O$ - 270 ppbv. Note (*): To account for the modelled climate variability for a 125 ka control scenario, we use an average of three 125 ka simulations which feature very minor GIS elevation changes between them (maximum scaling percentage of ±1.8 in Eq. 1) as our 125 ka control.

| Exp ID | NEEM $\Delta$z $(m)$ | Scaling % - $\beta$ | Sea Ice Forcing $(W/m^2)$ | Winter Sea Ice Extent $(10^6 km^2)$ |
|---|---|---|---|---|
| PI | 0 | 0 | 0 | 18.77 |
| 125Control * | 0 | 0 | 0 | 19.87 |
| p50 | 50 | +2 | 0 | 19.46 |
| p100 | 100 | +4 | 0 | 19.65 |
| p300 | 300 | +11 | 0 | 20.07 |
| p500 | 500 | +18 | 0 | 19.06 |
| p700 | 700 | +26 | 0 | 18.88 |
| p900 | 900 | +33 | 0 | 19.02 |
| p1100 | 1100 | +40 | 0 | 18.63 |
| p1300 | 1300 | +48 | 0 | 18.46 |
| m50 | -50 | -2 | 0 | 19.71 |
| m100 | -100 | -4 | 0 | 19.61 |
| m300 | -300 | -11 | 0 | 20.09 |
| m500 | -500 | -18 | 0 | 19.82 |
| m700 | -700 | -26 | 0 | 20.45 |
| m900 | -900 | -33 | 0 | 20.20 |
| m1100 | -1100 | -40 | 0 | 20.39 |
| m1300 | -1300 | -48 | 0 | 20.56 |
| GIS1-SIE-11.49 | 99 | - | 119.3 | 11.49 |
| GIS2-SIE-11.52 | -316 | - | 116 | 11.52 |
| GIS3-SIE-11.67 | 14 | - | 114.6 | 11.67 |
| GIS4-SIE-11.72 | 303 | - | 111.9 | 11.72 |
| GIS5-SIE-12.25 | 231 | - | 109.6 | 12.25 |

| Exp ID | NEEM $\Delta z$ ($m$) | Scaling percentage | Sea Ice Forcing ($W/m^2$) | Winter Sea Ice Extent ($10^6 km^2$) |
|---|---|---|---|---|
| GIS6-SIE-12.63 | -246 | - | 97.6 | 12.63 |
| GIS7-SIE-13.45 | -19 | - | 91.2 | 13.45 |
| GIS8-SIE-13.77 | -371 | - | 83.5 | 13.77 |
| GIS9-SIE-13.77 | -322 | - | 85.1 | 13.77 |
| GIS10-SIE-14.05 | -391 | - | 77.8 | 14.05 |
| GIS11-SIE-14.15 | 328 | - | 77.4 | 14.15 |
| GIS12-SIE-14.93 | -391 | - | 60.2 | 14.93 |
| GIS13-SIE-14.98 | 216 | - | 55 | 14.98 |
| GIS14-SIE-15.33 | -288 | - | 55.3 | 15.33 |
| GIS15-SIE-15.37 | -228 | - | 63 | 15.37 |
| GIS16-SIE-15.46 | -167 | - | 51 | 15.46 |
| GIS17-SIE-15.59 | -749 | - | 54.4 | 15.59 |
| GIS18-SIE-15.77 | -190 | - | 45.1 | 15.77 |
| GIS19-SIE-16.02 | -493 | - | 35.5 | 16.02 |
| GIS20-SIE-16.08 | -632 | - | 40 | 16.08 |
| GIS21-SIE-16.24 | -217 | - | 39.5 | 16.24 |
| GIS22-SIE-16.25 | 221 | - | 31.2 | 16.25 |
| GIS23-SIE-16.58 | -117 | - | 28.4 | 16.58 |
| GIS24-SIE-16.75 | -156 | - | 22.6 | 16.75 |
| GIS25-SIE-17.21 | -140 | - | 18.9 | 17.21 |
| GIS26-SIE-17.54 | -592 | - | 16.7 | 17.54 |
| GIS27-SIE-18.15 | -631 | - | 9.4 | 18.15 |
| GIS28-SIE-18.18 | -482 | - | 12.2 | 18.18 |
| GIS29-SIE-18.29 | -1053 | - | 7.5 | 18.29 |
| GIS30-SIE-18.65 | -343 | - | 8.4 | 18.65 |
| GIS31-SIE-19 | -719 | - | 0.9 | 19.00 |
| GIS32-SIE-19.48 | -292 | - | 1.3 | 19.48 |
| GIS1 | 99 | - | 0 | 19.14 |
| SIE-11.39 | 0 | - | 119.3 | 11.39 |
| GIS2 | -316 | - | 0 | 19.95 |
| SIE-11.83 | 0 | - | 116 | 11.83 |
| GIS13 | 216 | - | 0 | 19.52 |

| Exp ID | NEEM $\Delta$z ($m$) | Scaling percentage | Sea Ice Forcing ($W/m^2$) | Winter Sea Ice Extent ($10^6 km^2$) |
|---|---|---|---|---|
| SIE-15.65 | 0 | - | 55 | 15.65 |
| GIS31 | -719 | - | 0 | 19.97 |
| SIE-20.09 | 0 | - | 0.9 | 20.09 |

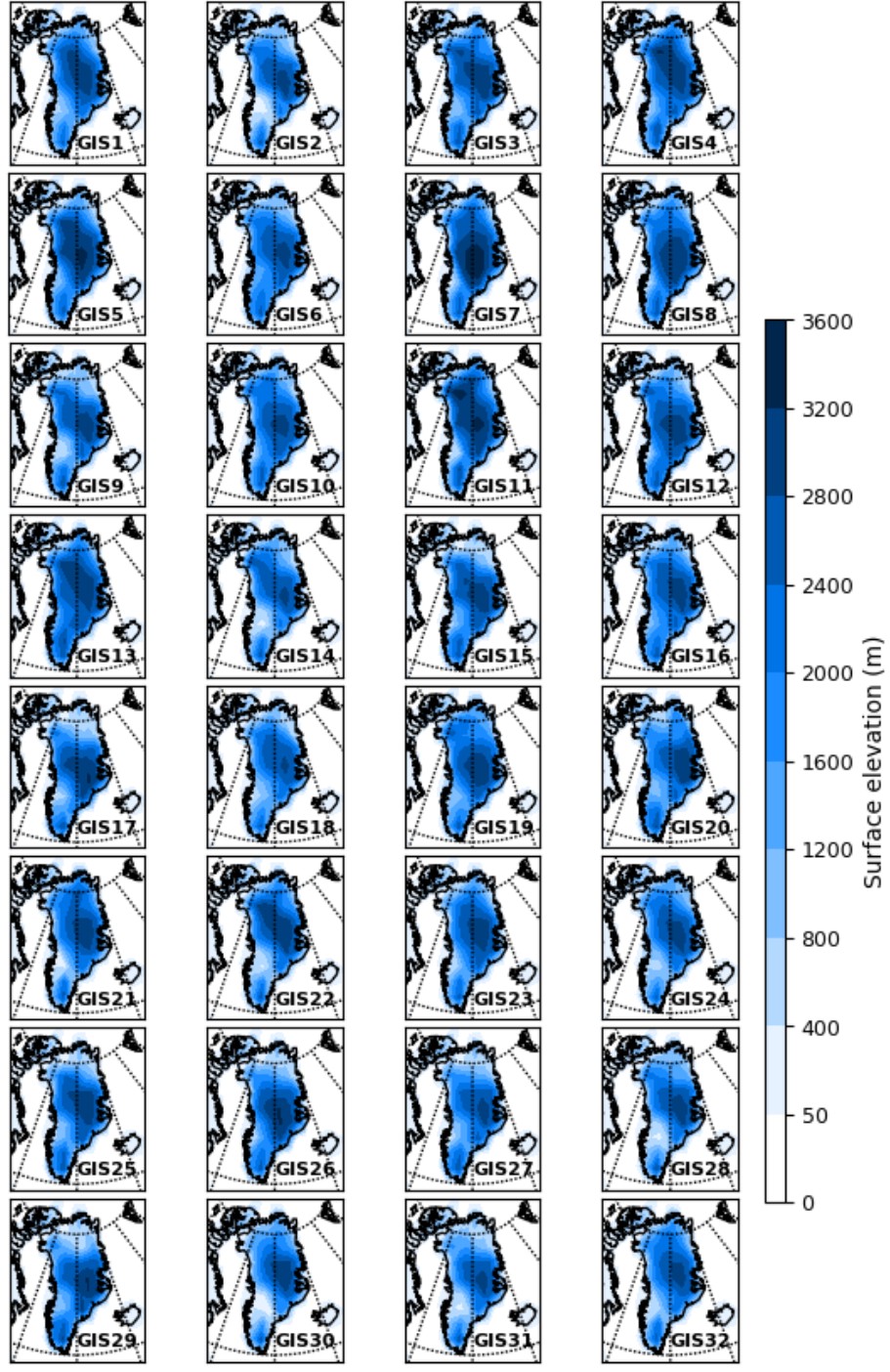

**Figure A1.** Surface elevation (meters) of LIG GIS morphologies used in the 32 simulations that examine the joint impact of modified Arctic sea ice retreat and modified GIS morphology. See Table. A1 for more details.

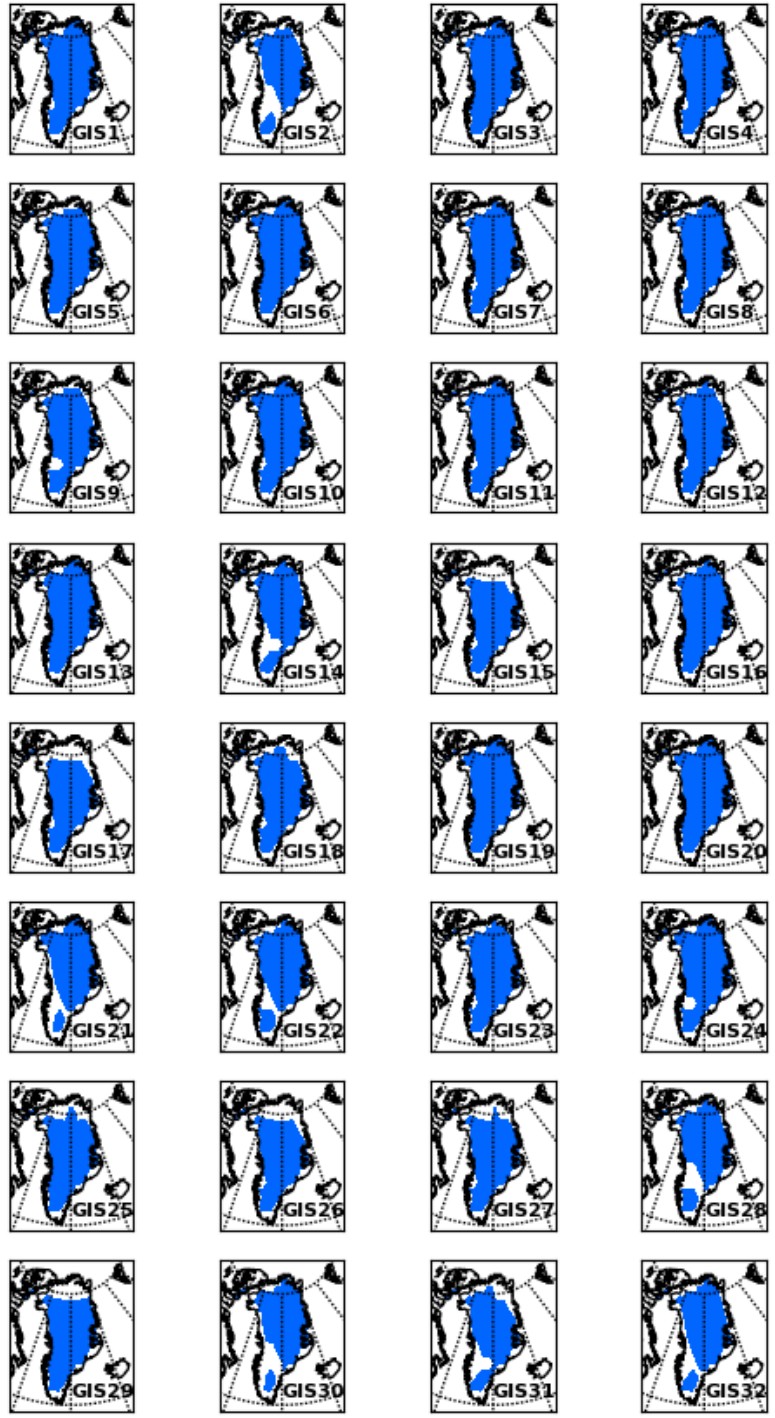

**Figure A2.** Land-Ice mask of LIG GIS morphologies used in the 32 simulations that examine the joint impact of modified Arctic sea ice retreat and modified GIS morphology. See Table. A1 for more details.

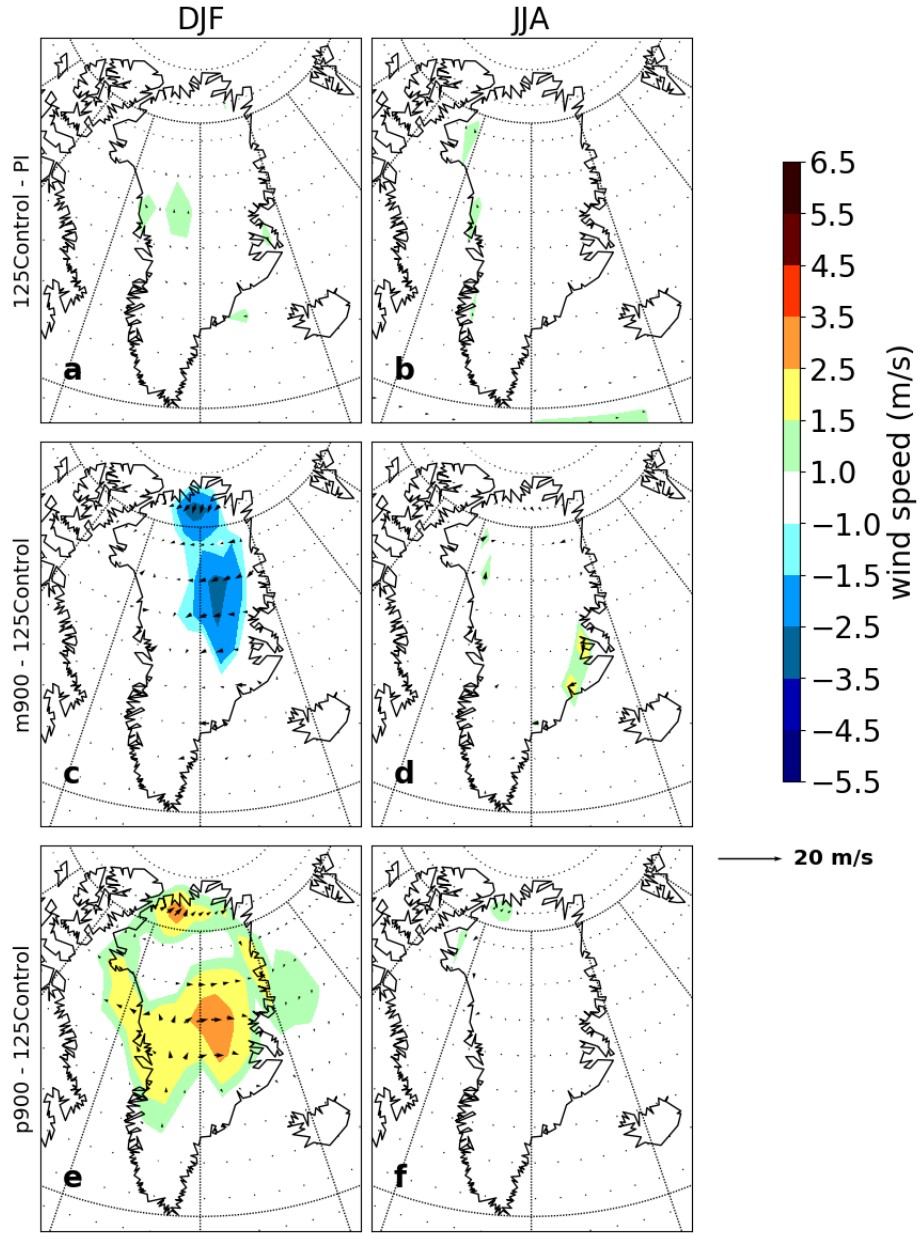

**Figure A3.** Winter (DJF) and summer (JJA) low-level winds (at 850 hPa) anomalies for m900 (c-d) and p900 (e-f) compared to the 125Control simulation. Also shown low-level wind anomalies for the 125Control compared to the PI simulation (a-b). Only the anomalies statistically significant at the 95% confidence level are displayed. Shading displays wind speed (m/s).

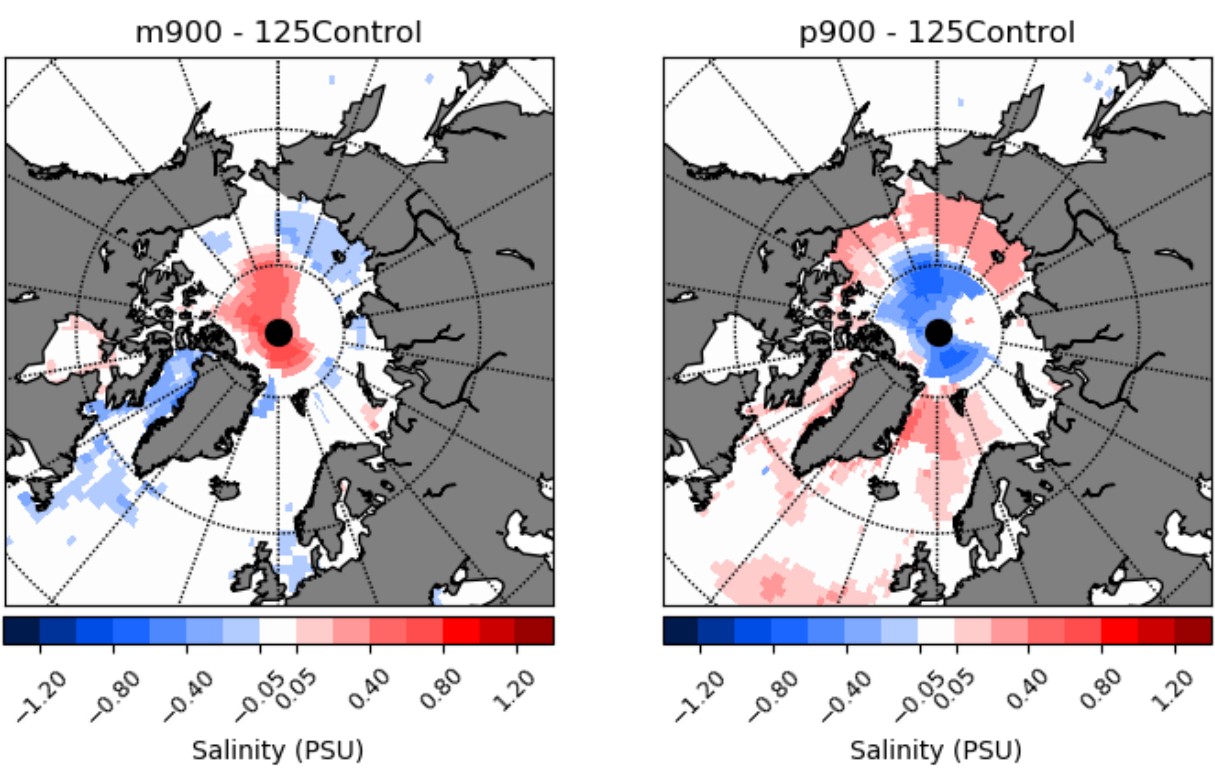

**Figure A4.** Annual salinity (PSU; averaged over the top 535 m) anomalies for m900 (a) and p900 (b) compared to 125Control simulation. Only the anomalies statistically significant at the 95% confidence level are displayed. Shading displays wind speed (m/s).

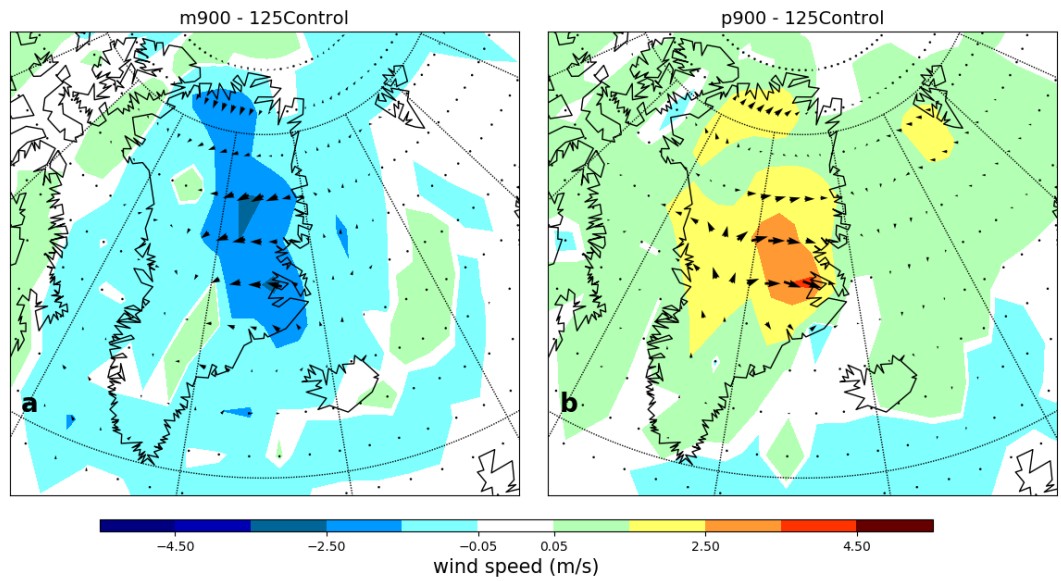

**Figure A5.** Change in 10 m winter (DJF) wind speed (in m/s) for (a) m900 and (b) p900 compared to 125Control. Only the anomalies statistically significant at the 95% confidence level are displayed.

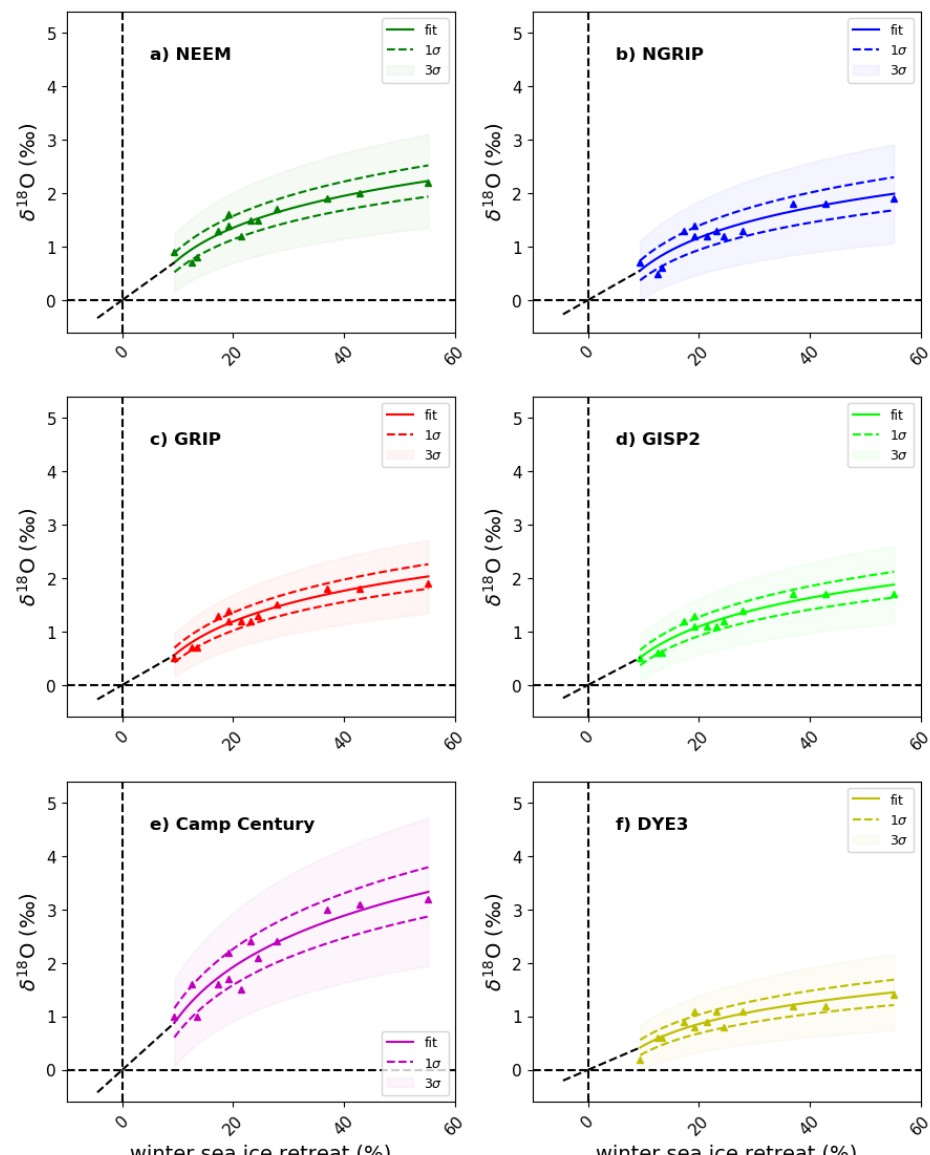

**Figure A6.** Simulated $\delta^{18}Op$ anomalies as a function of winter (March) sea ice retreat. The retreat of sea ice is calculated as the percentage change in winter sea ice extent compared to the 125 ka control simulation. Ice core sites shown: (a) NEEM, (b) NGRIP, (c) GRIP, (d) GISP2, (e) Camp Century, (f) DYE3. Triangles represent results from the sea ice sensitivity experiments performed by Malmierca-Vallet et al. (2018). Solid lines signify best fit lines (fit = b*(log(x) − a)). Also shown ±1s (lines with dashes) and ±3s uncertainty (shade envelopes) on the best fit curve. We assume a straight line regression (fit = a+bx) for winter sea ice losses between 0% and 9% and increases up to −4.5%. This figure is an adaptation of figure 10 shown in Malmierca-Vallet et al. (2018).

## Appendix B: The method of isolating the impacts of $\delta^{18}O$ due to sea ice changes

To calculate the sea-ice-corrected $\delta^{18}O$ anomalies we deduct the sea-ice-associated $\delta^{18}O$ effect from the total $\delta^{18}O$ anomalies (see section 3.2). In particular, we use the sea ice retreat simulations of Malmierca-Vallet et al. (2018) to isolate the impacts of $\delta^{18}O$ due to sea ice variation (Fig. A6). We acknowledge that this approach has its limitations as the interaction among GIS shape and sea ice factors could lead to smaller/larger effects than predicted from the sum of single parameter effects.

5  In order to test the robustness of our sea-ice-correction method, we run eight additional LIG simulations with the purpose of separating the effect of sea ice changes versus GIS shape changes. From the ensemble of 32 simulations, which explore the joint impacts of sea ice change and GIS change over Greenland, we select 4 simulations (GIS1-SIE-11.49, GIS2-SIE-11.52, GIS13-SIE-14.98 and GIS31-SIE-19) and rerun them with: (1) only the sea ice forcing implemented and, (2) only the modified GIS shape implemented (see Table. A1).

10  We find that the 4 simulations that explore the joint impact of GIS shape changes and sea ice changes result in smaller/larger $\delta^{18}Op$ anomalies (compared to the 125 ka control) than the predicted from the sum of single parameters effect (Table. B1 and B2). Nevertheless, differences are not higher than around ±1‰. This is within the model uncertainty of annual mean $\delta^{18}Op$ (Malmierca-Vallet et al., 2018).

Table B1: Simulated $\delta^{18}Op$ anomalies compared to 125 ka control on six Greenland deep ice cores: NEEM, NGRIP, GRIP, DYE3, GISP2 and Camp Century. For each pair of simulations, it is shown $\delta^{18}Op$ anomalies due to (1) GIS shape changes, (2) sea ice changes and, (3) sum of single parameter effects.

| Sum of single parameter effects | | | | | |
| --- | --- | --- | --- | --- | --- |
| **Exp ID** | **NEEM** | **NGRIP** | **GRIP** | **DYE3** | **GISP2** | **Camp Century** |
| **GIS1** | -0.9 | 0.0 | 0.2 | 2.1 | 0.6 | 0.4 |
| **SIE-11.39** | 1.3 | 1.5 | 1.6 | 0.5 | 1.4 | 2.0 |
| **SUM** | **0.4** | **1.4** | **1.8** | **2.6** | **1.9** | **2.5** |
| **GIS2** | 0.4 | 0.6 | 0.0 | 3.2 | 0.5 | -1.6 |
| **SIE-11.83** | 1.4 | 1.6 | 1.3 | 0.3 | 1.1 | 2.0 |
| **SUM** | **1.8** | **2.2** | **1.3** | **3.6** | **1.6** | **0.5** |
| **GIS13** | -1.8 | -0.7 | 1.1 | -1.2 | 1.2 | -2.1 |
| **SIE-15.65** | 0.6 | 1.1 | 1.1 | 0.2 | 0.9 | 0.6 |
| **SUM** | **-1.2** | **0.4** | **2.2** | **-1.0** | **2.1** | **-1.5** |
| **GIS31** | 2.5 | 1.5 | 0.4 | 3.1 | 0.9 | 1.2 |
| **SIE-20.09** | -0.6 | -0.4 | -0.3 | -0.7 | -0.4 | -1.2 |
| **SUM** | **1.9** | **1.2** | **0.1** | **2.4** | **0.5** | **0.0** |

Table B2: Simulated $\delta^{18}O$p anomalies compared to 125 ka control on six Greenland deep ice cores: NEEM, NGRIP, GRIP, DYE3, GISP2 and Camp Century. $\delta^{18}O$p anomalies due to the joint impact of GIS shape changes and sea ice changes.

| Simulated joint impact | | | | | | |
|---|---|---|---|---|---|---|
| Exp ID | NEEM | NGRIP | GRIP | DYE3 | GISP2 | Camp Century |
| GIS1-SIE-11.49 | 0.1 | 1.1 | 1.8 | 3.4 | 2.0 | 2.3 |
| GIS2-SIE-11.52 | 2.6 | 2.6 | 1.7 | 4.2 | 2.3 | 1.8 |
| GIS13-SIE-14.98 | -0.3 | 1.0 | 2.5 | -0.3 | 2.4 | 0.0 |
| GIS31-SIE-19 | 2.5 | 1.4 | 0.9 | 3.3 | 1.3 | 1.3 |

## Appendix C:  Model Evaluation

The performance of the atmospheric component of HadCM3 (HadAM3: Pope et al. (2000)) over Greenland, has been validated against the European Centre for Medium-Range Weather Forecasts (ECMWF) Re-Analysis (Murphy et al., 2002). They show reasonable agreement with temperature, precipitation and wind observations except for a small cold winter bias (associated with excessive longwave cooling), a warm summer bias (excessive shortwave heating at the surface), and a wet bias (related to inefficient orographic blocking) (Murphy et al., 2002). The performance of the HadCM3 coupled mode suffers from similar errors. In particular, the model's performance over the Greenland region and when coupled to the land surface scheme MOSES 2.1 has been published in Stone and Lunt (2013) and Valdes et al. (2017). There is generally good agreement with observed temperatures (derived from Hanna et al. (2005)) and observational annual precipitation (derived from Uppala et al. (2005)), except for a summer warm bias (1.9 °C) and an annual wet bias (1.4 mm/day) in southeast Greenland for both predicted and prescribed vegetation control runs (Stone and Lunt, 2013).

A validation of the isotope output has also been carried out for the atmosphere only (HadAM3; Sime et al. (2013)) as well as for the coupled ocean-atmosphere model (Tindall et al., 2009, 2010; Xinping et al., 2012). HadCM3 is able to reproduce the large-scale features of $\delta^{18}O$ in precipitation, covering altitude, latitude, amount and continental effects (Tindall et al., 2009). Moreover, Malmierca-Vallet et al. (2018) provides an evaluation of two control (PI and present-day experiments) HadCM3 isotope simulations over Greenland; HadCM3 shows similar heavy $\delta^{18}O$ biases over Greenland than other models (e.g., Sime et al., 2013; Sjolte et al., 2014).

Similar biases are expected to affect the PI and LIG experiments. To minimize the effect of the model bias over Greenland, and hence any influences on the study results, we use the standard approach and report modelled values as anomalies (LIG minus PI).

In addition, we note that the coarse spatial resolution of HadCM3 complicates to reliably model $\delta^{18}O$ changes at the coastal margins. Hence, the small Renland ice cap (where LIG ice has been retrieved) is not included in this study, as it is not well captured within the HadCM3 resolution.