# Peer review of "Sea-ice feedbacks influence the isotopic signature of Greenland Ice Sheet elevation changes: Last Interglacial HadCM3 simulations"

_Climate of the Past, 2020_

## Referee Comment (RC1) · Anonymous Referee #1 · 26 May 2020

Review

Irene Malmierca Vallet, Louise C. Sime, Paul J. Valdes, Julia C. Tindall

Sea-ice feedbacks influence the isotopic signature of Greenland Ice Sheet elevation changes: Last Interglacial HadCM3 simulations

This study investigates the changes of stable water isotopes in the precipitation over Greenland and their dependence on the height of the Greenland ice sheet. A set of climate simulations with the HadCM3 model is used for this purpose. The temporal

focus of the work is the climate of the Last Interglacial. The simulation results indicate non-linear complex processes in the hydrological cycle, which lead to different delta 18O-height relationships for a past increase or decrease of the current Greenland ice sheet height. Further analyses show that differences in Arctic winter sea ice cover seems to be the main reason for a significant change in the delta 18O-height relationship.

The results of the analyses are presented in a comprehensible order and discussed in detail. The manuscript is clearly structured and very well written, and the methodological approaches are comprehensible and insightful. Overall, I can recommend a publication of this paper in CP without hesitation and suggest only minor corrections and additions to the text.

Comments/Questions:

Page 3, Line7ff: What is the horizontal and vertical resolution of the HadCM3 simulation ensemble?

P3L21ff: The usage of the second simulation ensemble is less clearly described. No information of the change in GIS extent for the different simulations is found in the text or the appendix. It also remains unclear, how the different influence of GIS extent, GIS height and sea ice changes can be separated from this ensemble, as all three parameters are apparently changed at the same time.

P4L5ff: Some information about the overall (dis)agreement between modelled LIG-PI d18O anomalies as compared to ice core data should be added.

P4L5: A reference for the difference between NEEM drill site and deposition site should be given.

P4L9ff: References to some plots of Fig. 1 are wrong in this paragraph. E.g, NGRIP delta 18O changes are shown in Fig. 1e (not 1f), GISP2 values in Fig. 1m (not 1p), etc. Fig. 1z does not exist, at all.

P5L2: Are the differences in temperature lapse rate (0.47C/100m to 0.44C/100m) statistically significant or within the model-intrinsic uncertainty?

P5L6: Figure 5 is discussed before Figure 3+4 have been mentioned. This figure order could be improved.

P5L32: Is there an explanation for the non-linear behavior of precipitation changes?

P6L4ff: At Camp Century, the same sign in precipitation changes for increased and decreased elevation is explained by winter sea ice conditions in the Baffin Bay. The changes show in Fig.3d, 3f are only subtle and it is hard to believe that these minor differences have any effect on precipitation formation. They will barely change the high mean winter sea ice concentration of the 125k control simulation shown in Fig. 3e.

P6L11: For a better assessment of the simulated PI mean sea ice extent (5.8 mil km2), it should be compared to observed/reconstructed values.

P6L31: The reference to Fig. C1 would be better placed after the next sentence ("Since the surface [. . .] center of basin.").

P7L8: The method of isolating the impacts of delta 18O due to sea ice variation should be explained in more detail. E.g., how is a potential effect of changed GIS extent treated in this analysis. And why is this analysis performed for winter sea ice retreat, only? Are summer SIC changes negligible?

P8L26ff: In this paragraph simulated temperature lapse rates are compared to observed mean lapse rates, but also to changes in temperature lapse rates for a warming climate. These two quantities (mean state and its temporal deviation) should not be mixed in this comparison.

P9L21ff: How large is the modelled modern spatially derived delta 18O-elevation gradient? How does it compare to the cited values of Dansgaard/Johnsen et al./Vinther et al.?

Fig. 1: The last column of plots (Fig, 1d, 1h, etc.) lacks an explanation. Why is the plot of winter sea ice extent vs. GIS elevation changes different for the various ice core sites?

Figs. 2/3/5/6: Are all anomalies shown in the plots statistically significant? No non-significant values can be detected in these figures.

Fig. 3: The color bar values in the plots range from 0..1, but the figure caption states that sea ice concentration and its anomalies are given in percent (0..100%).

---

## Referee Comment (RC2) · Anonymous Referee #2 · 2 Jun 2020

Malmierca-Vallet and colleagues present simulations with an isotope-enabled climate model for the last interglacial (LIG), approximately 125,000 years before present. They analyze and discuss two ensembles of sensitivity simulations with artificially changed topography over the Greenland ice sheet (GIS) and different sea ice perturbations. Based on these simulations, the elevation gradients of the precipitation-weighted oxygen isotopic ratio d18Op are determined, quantified, and some feedbacks and nonlinearities are discussed.

The manuscript is written well and the figures adequately support the text. The topic

and findings are a very good match for Climate of the Past and the underlying questions are highly relevant for the interpretation of paleo proxy records. There are, however, a number of shortcomings that have to be addressed before publication. My most important criticism concerns the incomplete and sometimes confusing analysis of the simulations and, related to that, the incomplete discussion of the results as they compare to existing literature. A technical issue that needs to be addressed is that only insufficient details are included about the second ensemble of simulations. Most of the necessary information appears to be included in a paper under review (Domingo et al.). The full information on how this ensemble was constructed should be available to the readers of the present manuscript.

I was confused over how the two ensembles were used in the analysis. Ensemble 1 comprises 16 simulations where the only direct perturbation is the GIS topography. Other anomalies, including sea ice thickness and extent, are consequences thereof, mediated by the coupled model. Ensemble 2 perturbs the sea ice directly. However, since in the coupled model this is mostly a quantitative difference, not a qualitative one both ensembles could in principle be analyzed simultaneously throughout the paper and maybe this means that they should. For example, figure 7 shows results from all 48 simulations, but figure 1 only of the first ensemble.

Related to this, I believe the analysis falls short of answering the key question about which one of the perturbations, sea ice or elevation, has the stronger impact. Are there regional differences in the relative importance? Given the limited knowledge we have about sea ice and elevation anomalies during the LIG, how do the typical ranges of uncertainty in both variables translate into an unaccounted part of the reconstructed d18Op signal in ice cores? Are there regions that have a particularly large impact on d18Op and from which improved reconstructions would be especially insightful? I think the simulations presented here have the potential to make good progress on these important questions and answering some of them would greatly increase the interest and impact of this manuscript. I do not expect all of them answered, but a revised

manuscript should at least clarify the relative importance of sea ice and elevation per-
turbations more clearly.

The manuscript would greatly benefit from a more careful discussion of the existing
literature on the subject, in particular the three papers by Niklaus Merz from 2014 and
2016. The provide a very detailed analysis of how elevation changes impact temper-
ature over the GIS (2014a), precipitation (2014b), and on the role of sea ice (2016).
All three are referenced in Malmierca-Vallet et al. (2018), but only Merz et al. (2014b)
here. The dynamical consequences of an altered GIS topography are discussed in
detail in Merz et al. (2014a) and I think that several of the findings apply here, too. For
example, the strengthening of the Greenland anticyclone with higher elevations (Fig.
4) and the peripheral warming (Fig. 2; p8 l12ff) were found there too and turbulent
heat fluxes were the cause. This earlier paper also discusses changes in the seasonal
temperature cycle with some surprising details such as a marked cooling in regions
of lower topography in winter. I suspect that these effects are important for d18O and
should therefore be discussed here. The dynamic explanation that is currently given in
the manuscript is based on studies of glacial climate and how the Laurentide ice sheet
interacts with the atmosphere. The main effect described in these studies is how the
ice sheet barrier affects the jet stream and therefore the storm tracks. The Greenland
ice sheet is too far north to affect the storm tracks and hence the dynamics are very
different. Please remove these references as they are misleading (Felzer et al., 1996;
Singarayer and Valdes, 2010; Pausata et al., 2011). The (local) barrier effect is how-
ever very important for precipitation (e.g., p8 l6ff), as previously shown by Langen et
al. (2012), Hakuba et al (2012), and Merz et al. (2014b).

Lastly, HadCM3 is a model with a relatively low resolution, which is known to negatively
impact the circulation and surface climate over Greenland (Vizcaino et al, 2014). A
discussion of how this impacts the results should be included in the revised manuscript.
It is clear that the main advance of the present study is the inclusion of oxygen isotopes,
but the correct simulation of d18O depends on a good representation of the physical

climate system.

References: Langen et al. (2012): Self-inhibiting growth of the Greenland Ice, GRL 39, L12502, doi:10.1029/2012GL051810 Hakuba et al. (2012): Impact of Greenland's topographic height on precipitation and snow accumulation in idealized simulations, JOURNAL OF GEOPHYSICAL RESEARCH 117, D09107, doi:10.1029/2011JD017052, 2012 Vizcaino (2014): Ice sheets as interactive components of Earth System Models: progress and challenges, WIREs Clim Change 5, 557–568. doi: 10.1002/wcc.285 Irvali et al. (2020): A low climate threshold for south Greenland Ice Sheet demise during the late Pleistocene, PNAS 117, 190-195, doi: 10.1073/pnas.1911902116

########

Minor comments:

Reference list: The URL for Malmierca-Vallet et al., 2018 is incorrect. It contains "/doi.org" twice.

Methods: The idealized scaling for ensemble 1 strikes me as an odd choice and the naming convention is confusing. If I understood it correctly, the elevation of the entire ice sheet is scaled by a percentage. The naming is derived from the absolute anomaly in elevation at NEEM, which is not representative. Why not use the percentage as the name? Also, please include an explanation why this approach was chosen instead of the (arguably) more physical of Domingo et al. or Merz et al. (2014a,b).

p1 l20: Maybe Irvali et al. (PNAS, 2020) is a valuable addition?

p3 l4: prefer to use references to original work, not only IPCC AR4/5.

p3l25: A paper in review is not sufficient as a reference for a key method.

p4 l5ff: Since the increases and decreases in elevation are artificial, to what degree are the simulated d18O anomalies applicable to real-world reconstuctions? The

precipitation-weighted d18O will greatly depend on precipitation and therefore barrier effects (see above).

figure 1: Why use a piecewise linear fit? Nonlinearity is referred to in the text.

figure 2: Is there any sign of inversion in winter or other local temperature effects like in Merz et al.? How would this impact d18Op?

figure 4: Why not show anomalies like in all other figures?

p5 l10: typo: aN increase

p5 l15: This part would benefit from looking at the findings of Hakuba et al. (2012) and Merz et al. (2014a). Discuss why local dynamic and temperature changes are not seen in HadCM3 as compared to CCSM4.

p6 l6: "likely linked to the reduced winter sea ice" Isn't the 2nd ensemble there to test this hypothesis?

p6 l21: Pausata reference is on glacial climate and a change in elevation of the Laurentide ice sheet. Not relevant here.

p7 l8: "to isolate the effect of sea ice" How? Greenland topography changes here too.

figure 7: To my eye, sea ice correction does not really improve the fit. Overall, the "GIS" ensemble does not show a clear correlation between d18Op and ice core elevation. The cloud of points are rather round. Does the fit stem from all simulations or only the 1st ensemble? What would a fit (and std.dev.) for only the GIS ensemble look like? Is the sea ice correction straightforward? Does it not relate to the rather patchy (Fig. 6) precip anomalies? Is it not important in which region the sea ice reduction takes place?

p7 l24: I might have missed the definition, but I do not understand what exactly "core-average" means.

p7 l25: changes in gradient are described, followed by an argument that sea ice influences the linearity of the d18Op-elevation relationship. Gradients only describe the linear trend, so I am very confused over what this statement means. I think a non-linear metric, e.g., the curvature or similar, is needed.

p7 l29: "The dependence of the $\delta$ 18 O variable on elevation variations occurs in response to variations in winter sea ice extent." This can be interpreted as if the authors want to claim that elevation changes are a result of sea ice anomalies, and that the latter are the ultimate reason for isotope changes. This touches on an important point: The relative importance of sea ice and elevation changes and feedbacks between the two effects, that should be discussed in much more details. I think the two ensembles are well suited for such an analysis.

p8 l22: The combined changes discussed above are on the GIS ensemble, but figure 1 is on the elevation ensemble only. I think this is an error in the methodology.

section 4.3: This section compares modelled and observed isotopic lapse rates. It seems to me that the modelled lapse rates are derived from the first ensemble, i.e., it describes a change in d18Op as a function of the lower elevation at the same elevation, while the observed gradients have a spatial component, and thus a lower elevation generally implies a closer proximity to the coast. These should not be mixed or the difference must at least be clearly stated.

There is an excessive use of appendix figures for a journal that does not have very strict length restrictions.

---

## Author Comment (AC1) · 2 Sep 2020

**REVIEWER 1**

**Minor comments:**

**Minor comment 1:** Page 3, Line7ff: What is the horizontal and vertical resolution of the HadCM3 simulation ensemble?

**Response to minor comment 1:** In answer, the following text has been added to the manuscript to make this point clearer.

Page 3, Line 5: "The atmosphere component has a horizontal grid spacing of 2.5° (latitude) by 3.75° (longitude) and has 19 vertical levels (Gordon et al., 2000). The horizontal grid resolution of the ocean component is 1.25° by 1.25° with 20 vertical levels (Gordon et al., 2000)."

**Minor comment 2:** P3L21ff: The usage of the second simulation ensemble is less clearly described. No information of the change in GIS extent for the different simulations is found in the text or the appendix. It also remains unclear, how the different influence of GIS extent, GIS height and sea ice changes can be separated from this ensemble, as all three parameters are apparently changed at the same time.

**Response to minor comment 2:** Following the reviewer's comment, we have added a new figure (Fig. A2) in the appendix A, run new simulations, and provided additional information of the change in GIS extent for the different simulations in the main text. The new figure shows the GIS extent (land-ice mask) for each of the 32 simulations that examine the joint impact of modified Arctic sea ice retreat and modified GIS morphology. New text on simulations:

Page 4, Line 24: "The resulting 32 LIG GIS morphologies show strong variation in terms of both height and ice extent (Fig. A1 and Fig. A2). Some morphologies show a rather small retreat of the GIS, and others a possible division of the GIS into two domes, some display strong ice loss in the south, while others show substantial ice retreat in the north."

And on the reviewer's concerns on how to separate the influence of GIS extent, GIS height and sea ice changes:

The objective of the set of 32 simulations is to study the joint impacts of sea ice change and GIS change. To examine the influence of GIS height changes alone, we use the set of 16 simulations with idealised variations in the elevation of the GIS, where the present-day GIS extent (land-ice mask) is unmodified. Similarly, the sea ice retreat simulations of Malmierca-Vallet et al. (2018), which only include sea ice forcing changes, help to isolate the influence of $\delta^{18}O$ due to sea ice variations. The GIS extent is the only parameter whose effects cannot be isolated from any of our ensemble of simulations, which should be rather smaller compared to sea ice influences. We acknowledge that this approach has limitations as the interaction among the three factors (GIS elevation and extent and sea ice) could lead to smaller/larger effects than predicted from the sum of single parameter effects. Following the reviewer's suggestion, the authors have added a new appendix (Appendix B) that explores the robustness of this approach. We also run eight additional LIG simulations with the purpose of separating the effect of sea ice changes versus GIS shape changes.

Page 37, Appendix B: To calculate the sea-ice-corrected $\delta^{18}O$ anomalies we deduct the sea-ice-associated $\delta^{18}O$ effect from the total $\delta^{18}O$ anomalies (see section 3.2). In particular, we use the sea ice retreat simulations of Malmierca-Vallet (2018) to isolate the impacts of $\delta^{18}O$ due to sea ice variation (Fig. A6). We acknowledge that this approach has its limitations as the interaction among GIS shape and sea ice factors could lead to smaller/larger effects than predicted from the sum of single parameter effects. In order to test the robustness of our approach, we run eight additional LIG simulations with the purpose of separating the effect of sea ice changes versus GIS shape changes. From the ensemble of 32 simulations, which explore the joint impacts of sea ice change and GIS change over Greenland, we select 4 simulations (GIS1-SIE-11.49, GIS2-SIE-11.52, GIS13-SIE-14.98 and GIS31-SIE-19) and rerun them with: (1) only the sea ice forcing implemented and, (2) only the modified GIS shape implemented (see Table A1).

We find that the 4 simulations that explore the joint impact of GIS shape changes and sea ice changes result in smaller/larger $\delta^{18}O$ anomalies (compared to the 125 ka control) than the predicted from the sum of single parameters effect (Table. B1 and B2). Nevertheless, differences are not higher than around ±1‰. This is within the model uncertainty of annual mean $\delta^{18}Op$ (Malmierca-Vallet et al., 2018).

Table B1. Simulated $\delta^{18}Op$ anomalies compared to 125 ka control on six Greenland deep ice cores: NEEM, NGRIP, GRIP, DYE3, GISP2 and Camp Century. For each pair of simulations, it is shown $\delta^{18}Op$ anomalies due to (1) GIS shape changes, (2) sea ice changes and, (3) sum of single parameter effects

| Sum of single parameter effects | | | | | | |
|---|---|---|---|---|---|---|
| | **NEEM** | **NGRIP** | **GRIP** | **DYE3** | **GISP2** | **Camp Century** |
| **GIS1** | -0.9 | 0.0 | 0.2 | 2.1 | 0.6 | 0.4 |
| **SIE-11.39** | 1.3 | 1.5 | 1.6 | 0.5 | 1.4 | 2.0 |
| **SUM** | **0.4** | **1.4** | **1.8** | **2.6** | **1.9** | **2.5** |
| **GIS2** | 0.4 | 0.6 | 0.0 | 3.2 | 0.5 | -1.6 |
| **SIE-11.83** | 1.4 | 1.6 | 1.3 | 0.3 | 1.1 | 2.0 |
| **SUM** | **1.8** | **2.2** | **1.3** | **3.6** | **1.6** | **0.5** |
| **GIS13** | -1.8 | -0.7 | 1.1 | -1.2 | 1.2 | -2.1 |
| **SIE-15.65** | 0.6 | 1.1 | 1.1 | 0.2 | 0.9 | 0.6 |
| **SUM** | **-1.2** | **0.4** | **2.2** | **-1.0** | **2.1** | **-1.5** |
| **GIS31** | 2.5 | 1.5 | 0.4 | 3.1 | 0.9 | 1.2 |
| **SIE-20.09** | -0.6 | -0.4 | -0.3 | -0.7 | -0.4 | -1.2 |
| **SUM** | **1.9** | **1.2** | **0.1** | **2.4** | **0.5** | **0.0** |

Table B2: Simulated $\delta^{18}Op$ anomalies compared to 125 ka control on six Greenland deep ice cores: NEEM, NGRIP, GRIP, DYE3, GISP2 and Camp Century. $\delta^{18}Op$ anomalies due to the joint impact of GIS shape changes and sea ice changes.

| Simulated joint impact | | | | | | |
|---|---|---|---|---|---|---|
| | **NEEM** | **NGRIP** | **GRIP** | **DYE3** | **GISP2** | **Camp Century** |
| **GIS1-SIE-11.49** | 0.1 | 1.1 | 1.8 | 3.4 | 2.0 | 2.3 |
| **GIS2-SIE-11.52** | 2.6 | 2.6 | 1.7 | 4.2 | 2.3 | 1.8 |
| **GIS13-SIE-14.98** | -0.3 | 1.0 | 2.5 | -0.3 | 2.4 | 0.0 |

| GIS31-SIE-19 | 2.5 | 1.4 | 0.9 | 3.3 | 1.3 | 1.3 |

**Minor comment 3:** P4L5ff: Some information about the overall (dis)agreement between modelled LIG-PI d18O anomalies as compared to ice core data should be added.

**Response to minor comment 3:** Following the reviewer's suggestion, the authors have added the following discussion in the manuscript.

Page 11, Line 7: **Implications for NEEM $\delta^{18}$O and elevation reconstructions**

"Considering the NEEM elevation reconstruction, which indicates NEEM elevation differences of +45±350 m at 126 ka relative to present-day (NEEM community members, 2013), we find a most likely increase in $\delta^{18}$O values of between +0.8‰ to +3.5‰ relative to PI (Fig. 8). This relatively falls within the lower end of the uncertainty range of the reconstruction by Domingo et al. (2020): most likely LIG $\delta^{18}$O peak of +3.6‰ and uncertainty range between +2.7‰ to +4‰. The relatively small overlap between the $\delta^{18}$O record and the elevation reconstruction has already been discussed in Domingo et al (2020) and, could possibly reflect uncertainties attached to the air content NEEM elevation reconstruction method. The methodology depends on making corrections to air content measurements related to insolation and temperature in conjunction with secular variations in surface pressure and winds (Raynaud et al., 2007; Martinerie et al., 1994; Krinner et al., 2000; Eicher et al., 2016). In addition, NEEM air content measurements between 127 and 118.3 ka are known to be affected by surface melting (NEEM community members, 2013)."

**Minor comment 4:** P4L5: A reference for the difference between NEEM drill site and deposition site should be given.

**Response to minor comment 4:** In answer, the following reference has been added to the manuscript to make this point clearer.

Page 5, Line 5: "NEEM community members (2013)"

**Minor comment 5:** P4L9ff: References to some plots of Fig. 1 are wrong in this paragraph. E.g, NGRIP delta 18O changes are shown in Fig. 1e (not 1f), GISP2 values in Fig. 1m (not 1p), etc. Fig. 1z does not exist, at all.

**Response to minor comment 5:** Corrected.

**Minor comment 6:** P5L2: Are the differences in temperature lapse rate (0.47C/100m to 0.44C/100m) statistically significant or within the model-intrinsic uncertainty?

**Response to minor comment 6:** In answer, the following text has been added to the manuscript.

Page 6, line 7: "Averaging across six ice core sites (Camp Century, NEEM, NGRIP, GRIP, GISP2 and DYE3), temperature lapse rates vary slightly from 0.47°C per 100 m for the lowered GIS states to 0.44°C per 100 m for the enlarged GIS states (Fig. 1), however these changes are not statistically significant."

**Minor comment 7:** P5L6: Figure 5 is discussed before Figure 3+4 have been mentioned. This figure order could be improved.

**Response to minor comment 7:** Done.

**Minor comment 8:** P5L32: Is there an explanation for the non-linear behavior of precipitation changes?

**Response to minor comment 8:** Precipitation is a highly non-linear phenomena; elevation is one of the main topo-climatic drivers of precipitation gradients, however the relationship between elevation and precipitation can be idiosyncratic (Spreen, 1947; Basist et al., 1994; Daly, 1994). In many regions including over the GIS, local changes in precipitation with elevation can approximate a curved distribution and best estimated by non-linear models (e.g. Marquínez et al., 2003; Körner, 2007; van de Berg et al., 2013).

**Minor comment 9:** P6L4ff: At Camp Century, the same sign in precipitation changes for increased and decreased elevation is explained by winter sea ice conditions in the Baffin Bay. The changes show in Fig.3d, 3f are only subtle and it is hard to believe that these minor differences have any effect on precipitation formation. They will barely change the high mean winter sea ice concentration of the 125k control simulation shown in Fig. 3e.

**Response to minor comment 9:** We agree with the reviewer that changes in winter sea ice in the Baffin Bay are rather small and therefore, have probably minor effects on precipitation formation. We propose a new argument; the different behaviour found at Camp Century site is likely linked to its coastal position. Slope areas receive more moist air masses which are orographically lifted and consequently condensate and precipitate.

Page 7, Line 13: At Camp Century site, precipitation tends to increase with increasing surface elevation (-0.091 mm/year per 100 m elevation increase) (Fig. 1w). Removing the Camp Century site increases the core-average precipitation gradient to 0.029 mm/year per 100 m elevation increase. The different behaviour found at Camp Century site is likely linked to the orographic enhancement of precipitation (Johnson and Hanson, 1995; Frei and Schär, 1998; Petersen et al., 2004; Roe and Bakker, 2006) among the enlarged GIS states. Reductions in Camp Century height results also in marginal increases in precipitation rate (0.05 mm/year per 100 m) which are probably related to the weakening of the Greenland anticyclone and smaller barrier effect.

**Minor comment 10:** P6L11: For a better assessment of the simulated PI mean sea ice extent (5.8 mil km2), it should be compared to observed/reconstructed values.

**Response to minor comment 10:** In answer, the following text has been added to the manuscript to make this point clearer.

Page 3, Line 9: "The HadCM3 sea ice output over the Arctic Ocean has been previously validated against observational sea ice data (Meier et al., 2017; Peng et al., 2013) by Malmierca-Vallet et al. (2018). Under PI conditions, HadCM3 simulates rather little summer sea ice over the central Arctic Ocean, and too much winter sea ice over the Norwegian, Barents, Labrador and Bering Seas. For a full validation of the sea ice model, interested readers are referred to Malmierca-Vallet et al. (2018) and Gordon et al. (2000)."

**Minor comment 11:** P6L31: The reference to Fig. C1 would be better placed after the next sentence ("Since the surface [: : :] center of basin.").

**Response to minor comment 11:** Done.

**Minor comment 12:** P7L8: The method of isolating the impacts of delta 18O due to sea ice variation should be explained in more detail. E.g., how is a potential effect of changed GIS extent treated in this analysis. And why is this analysis performed for winter sea ice retreat, only? Are summer SIC changes negligible?

**Response to minor comment 12:** We have added a new Appendix B that explores the robustness of the method used to isolate the impacts of $\delta^{18}$O due to sea ice changes. See also response to minor comment 2.

We performed this analysis for winter sea ice retreat because summer sea ice changes tend to be smaller and present more noise than changes in winter sea ice. In order to test our method, we also calculate $\delta^{18}$O anomalies (compared to 125 ka control) corrected for annual sea ice changes. In particular, we use the sea ice retreat simulations of Malmierca-Vallet et al. (2018) to isolate the impacts of $\delta^{18}$O due to annual sea ice variation; we calculate the change in $\delta^{18}$O as a function of annual sea ice retreat.

To calculate $\delta^{18}$O anomalies corrected for annual sea ice changes, we deduct the annual-sea-ice-associated $\delta^{18}$O effect from the total $\delta^{18}$O anomalies. The figure above is very similar to Fig. 9 of the main paper. It shows $\delta^{18}$O anomalies as a function of the ice core site elevation change (m) relative to: (1) PI (red fit), (2) 125 ka control (blue fit), (3) winter-sea-ice corrected $\delta^{18}$O anomalies relative to 125 ka control (purple fit) and, (4) annual-sea-ice corrected $\delta^{18}$O anomalies relative to 125 ka control (black dashed fit). It is evident that the resulting $\delta^{18}$O anomalies corrected for both annual and winter sea ice changes are very similar, almost identical (Figure above – purple curve fit and black dashed curve fit).

[Figure]

This figure is identical to Fig.9 of the main paper but adding a fourth fit; annual-sea-ice corrected δ¹⁸O anomalies relative to 125 ka control (black dashed fit).

**Minor comment 13:** P8L26ff: In this paragraph simulated temperature lapse rates are compared to observed mean lapse rates, but also to changes in temperature lapse rates for a warming climate. These two quantities (mean state and its temporal deviation) should not be mixed in this comparison.

**Response to minor comment 13:** Following the reviewer's suggestion, the following text has been removed from the manuscript.

Page 9: "Our results are also in agreement with Erokhina et al., 2017, who point to a non-stationarity response of the climate to GIS elevation changes during the Holocene and Last Glacial Maximum (LGM). Erokhina et al., 2017 propose that following the transition from the LGM to the Holocene, mean annual temperature lapse rates over the GIS decreased by almost 20 %."

**Minor comment 14**: P9L21ff: How large is the modelled modern spatially derived delta 18O-elevation gradient? How does it compare to the cited values of Dansgaard/Johnsen et al./Vinther et al.?

**Response to minor comment 14:** In answer, the following text has been modified/added to the manuscript.

Page 10, Line 30: "Interestingly, LIG isotopic lapse rates and the PI spatially derived isotopic lapse (0.37‰ per 100 m) modelled with HadCM3, are lower than the modern spatially derived gradients of 0.62‰ per 100 m and 0.72‰ per 100 m in central and northwest Greenland respectively (Dansgaard et al., 1973; Johnsen et al., 1989; Vinther et al., 2009)."

Page 11, Line 5: "The HadCM3 resolution does not permit it to represent the steep GIS margins; this may be behind some of the model-data mismatches (Toniazzo et al., 2004)."

**Minor comment 15:** Fig. 1: The last column of plots (Fig, 1d, 1h, etc.) lacks an explanation. Why is the plot of winter sea ice extent vs. GIS elevation changes different for the various ice core sites?

**Response to minor comment 15:** In answer, the following text has been added to the manuscript to make this point clearer.

Page 19: In the last column of plots, winter sea ice extent vs GIS elevation changes differs for the various ice core sites because of the different elevation changes at each ice core site compared to the 125 ka control.

**Minor comment 16:** Figs. 2/3/5/6: Are all anomalies shown in the plots statistically significant? No nonsignificant values can be detected in these figures.

**Response to minor comment 16**: All anomalies shown in the plots are statistically significant at the 95% confident level.

**Minor comment 17:** Fig. 3: The color bar values in the plots range from 0..1, but the figure caption states that sea ice concentration and its anomalies are given in percent (0..100%).

**Response to minor comment 17:** In answer, we have changed the bar values in the plots which range now from 0% to 100%. See figure below:

[Figure]

Updated Fig. 5 in revised manuscript.

**REFERENCES**

Basist, A., Bell, G. D. & Meentemeyer, V. Statistical Relationships between Topography and Precipitation Patterns. *J. Clim* 7, 1305–1315 (1994).

Daly, C., Neilson, R. P. & Phillips, D. L. A Statistical-Topographic Model for Mapping Climatological Precipitation over Mountainous Terrain. *J. Appl. Meteorol.* 33, 140–158 (1994).

Körner, C. The use of 'altitude' in ecological research. *Trends Ecol. Evol.* 22, 569–574 (2007).

Marquínez, J., Lastra, J., García, P. Estimation models for precipitation in mountainous regions: the use of GIS and multivariate analysis. J. of Hydrology, 270, 1-11, 2003. doi:10.1016/S0022-1694(02)00110-5.

Spreen, W. C. A determination of the effect of topography upon precipitation. *Eos Trans. Am. Geophys. Union* 28, 285-290 (1947).

Van de Berg, W.J., van den Broeke, M. R., van Meijgaard, E., Kaspar, F. Importance of precipitation seasonality for the interpretation of Eemian ice core isotope records from Greenland. Climate of the Past, 9, 1589-1600 (2013). doi:10.5194/cp-9-1589-2013.

---

## Author Comment (AC2) · 3 Sep 2020

**Major comments:**

**Major comment 1:** My most important criticism concerns the incomplete and sometimes confusing analysis of the simulations and, related to that, the incomplete discussion of the results as they compare to existing literature. A technical issue that needs to be addressed is that only insufficient details are included about the second ensemble of simulations. Most of the necessary information appears to be included in a paper under review (Domingo et al.). The full information on how this ensemble was constructed should be available to the readers of the present manuscript.

**Response to major comment 1:** Following the reviewer's suggestion, we have added a new subsection (Experimental setup; Joint impact of changes in sea ice and GIS morphology) and a new figure (Fig A2) which provides additional information about the second ensemble of simulations. Also, the paper Domingo et al. (2020) is now published in Journal of Geophysical Research: Earth Surface, 125, 1-19, doi:10.1029/2019JF005237.

The following text has been added:

Page 4, Line 8: "The parameterisation of the set of GIS morphologies and sea ice retreat scenarios is performed by means of a Principal Component Analysis (PCA) approach. Due to the spherical geometry of the Earth, the application of a classical PCA to our data would be inappropriate. We thus apply a particular case of generalised PCA analysis (weighted PCA), described in Jollie et al (2002) and Salter et al. (2019)."

Page 4, Line 15: "In particular, we generate a set of 32 nine-dimensional random vectors. The first eight components of each vector are independently normally distributed and are used to generate a new GIS morphology via linear combination of the PCs (same procedure as in Domingo et al. (2020)). The ninth component represents instead heat flux, and is uniformly distributed between 0 and 120 W/m$^2$. We follow the methodology of Holloway et al. (2016, 2017) and Malmierca-Vallet et al. (2018) on sea ice forcing and implement these heat fluxes to the bottom of the Arctic sea ice (see table A1). This sea ice forcing is kept constant through the complete annual cycle, and thus the model still calculates the seasonal cycle of sea ice growth and decay. Sea ice varies over time with the coupled model, and both the oceanic and atmospheric components of HadCM3 respond to variations in sea ice (for more details on the methodology, see Malmierca-Vallet et al. 2018)."

Page 4, Line 24: "The resulting 32 LIG GIS morphologies show strong variation in terms of both height and ice extent (Fig. A1 and Fig. A2). Some morphologies show a rather small retreat of the GIS, and others a possible division of the GIS into two domes, some display strong ice loss in the south, while others show substantial ice retreat in the north."

Page 4, Line 27: "After the design GIS morphologies and associated sea ice forcing are generated, HadCM3 is used to model the isotopic response to these modified GIS morphologies and sea ice retreat scenarios at the ice-core sites. All LIG experiments are forced with orbital parameters and GHG values appropriate for 125 ka and time integrated for a total of 475-years."

**Major comment 2:** I was confused over how the two ensembles were used in the analysis. Ensemble 1 comprises 16 simulations where the only direct perturbation is the GIS topography. Other anomalies, including sea ice thickness and extent, are consequences thereof, mediated by the coupled model. Ensemble 2 perturbs the sea ice directly. However, since in the coupled model this is mostly a quantitative difference, not a qualitative one both ensembles could in principle be analyzed simultaneously throughout the paper and maybe this means that they should. For example, figure 7 shows results from all 48 simulations, but figure 1 only of the first ensemble.

Related to this, I believe the analysis falls short of answering the key question about which one of the perturbations, sea ice or elevation, has the stronger impact. Are there regional differences in the relative importance? Given the limited knowledge we have about sea ice and elevation anomalies during the LIG, how do the typical ranges of uncertainty in both variables translate into an unaccounted part of the reconstructed d18Op signal in ice cores? Are there regions that have a particularly large impact on d18Op and from which improved reconstructions would be especially insightful? I think the simulations presented here have the potential to make good progress on these important questions and answering some of them would greatly increase the interest and impact of this manuscript. I do not expect all of them answered, but a revised manuscript should at least clarify the relative importance of sea ice and elevation perturbations more clearly.

**Response to major comment 2:** In answer, the only perturbation in the first set of simulations is the GIS height. Changes in other variables, including the sea ice concentration and extent, are an output of the model. The second set of simulations modifies the sea ice directly; we implement heat fluxes (from 0.9 to 119.3 $W/m^2$) to the bottom of the Arctic sea ice. This sea ice forcing is kept constant thought the whole annual cycle, and the model still calculates the seasonal cycle of sea ice growth and decay. Sea ice varies over time with the coupled model, and both the oceanic and atmospheric components of HadCM3 respond to variations in sea ice. This makes the two set of simulations comparable.

The first part of the manuscript is focused on studying the solely impact of GIS elevation changes on Greenland $\delta^{18}O$ and underlying process. The second part of the manuscript is used to test to what extent the background climate state (sea ice changes) may influence the isotopic lapse rate values over Greenland. This is the main reason why Fig.1 shows results of only the 16 GIS elevation change simulations and Fig. 7 results from all 48 simulations.

Following the reviewer's suggestion, the authors have added two new sections with the titles "**Implications for NEEM $\delta^{18}O$ and elevation reconstructions**" and "**Relative influence of sea ice and GIS changes on ice core $\delta^{18}O$**". In this two sections, we: (1) compare the model results with the NEEM $\delta^{18}O$ and elevation reconstructions, and (2) we provide information about the relative importance of sea ice and ice sheet changes on determining $\delta^{18}O$ changes at each ice core site. See also response to minor comment 3 (reviewer 1) for the text of the section "Implications for the NEEM $\delta^{18}O$ and elevation reconstructions".

**Page 11, Line 18: Relative influence of sea ice and GIS changes on ice core $\delta^{18}O$**

"Considering the maximum reduction in NEEM's surface elevation proposed by the NEEM community members. (2013) of -305 m at 126 ka, we find that the impact of LIG orbital-sea ice changes appears to be the dominant factor determining $\delta^{18}O$ changes (explaining 60% of

the $\delta^{18}O$ anomaly), followed by GIS-driven sea ice changes (Fig. 9a). This is in agreement with previous studies that show the importance of changes in GIS topography and sea ice retreat to explain the LIG warming at the NEEM ice core site (Merz et al., 2014a, 2016; Guarino et al. 2020).

To make a comparable analysis at the other ice core sites, of the relative influence of each factor on determining $\delta^{18}O$ changes, we consider the same reduction in surface elevation of -305 m at the other locations (Fig. 9). We find that LIG orbital-sea ice changes is the dominant factor determining $\delta^{18}O$ changes at NGRIP, GRIP and GISP2 (accounting for 55-58% of the $\delta^{18}O$ changes) (Fig. 9b-d), while ice sheet changes appears to have the largest impact on $\delta^{18}O$ changes at DYE3 site (accounting for 48% of the $\delta^{18}O$ anomaly) (Fig. 9f). The highest sea ice influence is found at Camp Century (explain 10% of the $\delta^{18}O$ changes) (Fig. 9e).

Note, the above-mentioned relative influence of each parameter on $\delta^{18}O$ changes should be interpreted with caution; these results could change substantially if we were to consider any other possible elevation change scenario. There is no independent gas content information on elevation changes for DYE3 and Camp Century. Moreover, although there is total air content records that were measured on the GRIP (Raynaud et al., 1997) and NGRIP (Eicher et al. 2016) ice cores, the authors show how complex it is to interpret this proxy in term of elevation changes at the drilling site.

Additional data on elevation changes together with better dated ice, especially at DYE3 and Camp Century sites, would be particularly valuable to further assess our quantitative elevation change scenarios. In addition, considering sea ice and ice changes in a joint framework following a Gaussian Process emulation approach (Domingo et al., 2020) and take also account of isostatic change may permit a valuable quantitative assessment of how changes in the GIS affected LIG global sea levels."

**Major comment 3:** The manuscript would greatly benefit from a more careful discussion of the existing literature on the subject, in particular the three papers by Niklaus Merz from 2014 and 2016. The provide a very detailed analysis of how elevation changes impact temperature over the GIS (2014a), precipitation (2014b), and on the role of sea ice (2016). All three are referenced in Malmierca-Vallet et al. (2018), but only Merz et al. (2014b) here. The dynamical consequences of an altered GIS topography are discussed in detail in Merz et al. (2014a) and I think that several of the findings apply here, too. For example, the strengthening of the Greenland anticyclone with higher elevations (Fig. 4) and the peripheral warming (Fig. 2; p8 l12ff) were found there too and turbulent heat fluxes were the cause. This earlier paper also discusses changes in the seasonal temperature cycle with some surprising details such as a marked cooling in regions of lower topography in winter. I suspect that these effects are important for d18O and should therefore be discussed here. The dynamic explanation that is currently given in the manuscript is based on studies of glacial climate and how the Laurentide ice sheet interacts with the atmosphere. The main effect described in these studies is how the ice sheet barrier affects the jet stream and therefore the storm tracks. The Greenland ice sheet is too far north to affect the storm tracks and hence the dynamics are very different. Please remove these references as they are misleading (Felzer et al., 1996; Singarayer and Valdes, 2010; Pausata et al., 2011). The (local) barrier effect is however very important for precipitation (e.g., p8 l6ff), as previously shown by Langen et al. (2012), Hakuba et al (2012), and Merz et al. (2014b).

**Response to major comment 3:** Following the reviewer's suggestion, we have revised our discussion of the existing literature on the subject and now explicitly mention the three papers by Niklaus Merz from 2014 and 2016 and have removed the following references: Felzer et al., 1996; Singarayer and Valdes, 2010; Pausata et al., 2011. In addition, we examine temperature inversion and barrier effects in more detail in the revised manuscript.

Page 6, Line 1: "A previous study by Merz, et al. (2014a) found that the sensitivity of LIG Greenland's climate to GIS topography changes is seasonally diverse. For example, in winter, strong cooling conditions are found over some areas that become ice-free and flat, while the remaining ice dome show warmer conditions (Merz et al., 2014a). Areas that become ice-free are characterised by weak surface winds and turbulence, barring an efficient sensible heat flux and leading to very strong temperature inversion (Merz et al. 2014a). Our idealised elevation change simulations do not show this temperature inversion mechanism; this is most likely linked to the unchanged land-ice distribution in our experiments."

Page 9, Line 11: "Support comes also from Merz et al. (2014a) who show, for their perturbed LIG experiments with reduced GIS, a smaller anticyclone as well as decreased wind velocities."

Page 9, Line 14: "These modelled results are in agreement with the findings of Merz et al. (2014) and Hakuba et al. (2012) who show that a decrease in the height and size of the GIS weakens the barrier effect, permitting more moisture to be advected to the plateau."

Page 9, Line 22: "Similar mechanisms causing this surface temperature pattern were discussed in Merz et al. (2014a) who investigated the sensitivity of LIG Greenland's climate to GIS topography changes. In particular, Merz et al. (2014a) found cooling over areas of higher elevation (eastern Greenland) but warming on the periphery of the ice sheet. GIS topography changes influence the Greenland's surface energy balance through changes in surface winds and turbulent heat fluxes (Merz et al., 2014a)."

Page 11, Line 22: "This is in agreement with previous studies that show the importance of changes in GIS topography and sea ice retreat in the Nordic seas to explain the LIG warming at the NEEM ice core site (Merz et al., 2014a, 2016)."

**Major comment 4:** Lastly, HadCM3 is a model with a relatively low resolution, which is known to negatively impact the circulation and surface climate over Greenland (Vizcaino et al, 2014). A discussion of how this impacts the results should be included in the revised manuscript. It is clear that the main advance of the present study is the inclusion of oxygen isotopes, but the correct simulation of d18O depends on a good representation of the physical climate system.

**Response to major comment 4:** We agree with the reviewer that the HadCM3 resolution may be behind some of the model-data mismatches. Following the reviewer's suggestion, we have included an evaluation of the model's representation of the physical climate system over Greenland.

Page 39: **Appendix C - Model evaluation**

The performance of the atmospheric component of HadCM3 (HadAM3: Pope et al. (2000)) over Greenland, has been validated against the European Centre for Medium-Range Weather Forecasts (ECMWF) Re-Analysis (Murphy et al., 2002). They show reasonable agreement with

temperature, precipitation and wind observations except for a small cold winter bias (associated with excessive longwave cooling), a warm summer bias (excessive shortwave heating at the surface), and a wet bias (related to inefficient orographic blocking) (Murphy et al., 2002). The performance of the HadCM3 coupled mode suffers from similar errors. In particular, the model's performance over the Greenland region and when coupled to the land surface scheme MOSES 2.1 has been published in (Stone and Lunt, 2013) and (Valdes et al., 2017). There is generally good agreement with observed temperatures (derived from Hanna et al. (2005)) and observational annual precipitation (derived from Uppala et al. (2005)), except for a summer warm bias (1.9 °C) and an annual wet bias (1.4 mm/day) in southeast Greenland for both predicted and prescribed vegetation control runs (Stone and Lunt, 2013).

A validation of the isotope output has also been carried out for the atmosphere only (HadAM3; Sime et al. (2013)) as well as for the coupled ocean-atmosphere model (Tindall et al., 2009, 2010; Xinping et al., 2012). HadCM3 is able to reproduce the large-scale features of $\delta^{18}O$ in precipitation, covering altitude, latitude, amount and continental effects (Tindall et al., 2009). Moreover, Malmierca-Vallet et al. (2018) provides an evaluation of two control (PI and present-day experiments) HadCM3 isotope simulations over Greenland; HadCM3 shows similar heavy $\delta^{18}O$ biases over Greenland than other models (e.g., Sime et al., 2013; Sjolte et al., 2014).

Similar biases are expected to affect the PI and LIG experiments. To minimize the effect of the model bias over Greenland, and hence any influences on the study results, we use the standard approach and report modelled values as anomalies (LIG minus PI).

In addition, we note that the coarse spatial resolution of HadCM3 complicates to reliably model $\delta^{18}O$ changes at the coastal margins. Hence, the small Renland ice cap (where LIG ice has been retrieved) is not included in this study, as it is not well captured within the HadCM3 resolution.

Page 11, Line 5: ""The HadCM3 resolution does not permit it to represent the steep GIS margins; this may be behind some of the model-data mismatches (Toniazzo et al., 2004)."

**Minor comments:**

**Minor comment 1:** Reference list: The URL for Malmierca-Vallet et al., 2018 is incorrect. It contains "/doi.org" twice.

**Response to minor comment 1:** Done.

**Minor comment 2:** Methods: The idealized scaling for ensemble 1 strikes me as an odd choice and the naming convention is confusing. If I understood it correctly, the elevation of the entire ice sheet is scaled by a percentage. The naming is derived from the absolute anomaly in elevation at NEEM, which is not representative. Why not use the percentage as the name? Also, please include an explanation why this approach was chosen instead of the (arguably) more physical of Domingo et al. or Merz et al. (2014a,b).

**Response to minor comment 2:** The reviewer is correct that the entire GIS elevation is scaled up/down by a percentage. We add a new column in Table. A1 with the scaling percentage to provide these figures. However naming with the absolute change in elevation at NEEM is arguably more useful since it eases comparison of implemented elevation changes between

the first and second set of simulations. This is because the latter do not include scaling percentage.

Text changes:

Page 3, line 26: "The 16 idealised simulations are identical except for the GIS elevation, which is decreased/increased by ±2% up to ±48% (β scaling percentage – Table. A1). This decrease/increase is applied evenly over the entire GIS. This simple method is used because it shows a well controlled and more comprehensible idealised framework for sensitivity studies about the dependence of $\delta^{18}O$ and climate on the magnitude and sign of GIS elevation changes."

**Minor comment 3:** p1 l20: Maybe Irvali et al. (PNAS, 2020) is a valuable addition?

**Response to minor comment 3:** Done.

**Minor comment 4:** p3 l4: prefer to use references to original work, not only IPCC AR4/5.

**Response to minor comment 4:** Done.

**Minor comment 5:** p3l25: A paper in review is not sufficient as a reference for a key method.

**Response to minor comment 5:** The paper of Domingo et al. (2020) is now published in Journal of Geophysical Research: Earth Surface, 125, 1-19, doi:10.1029/2019JF005237. See also response to major comment 1.

**Minor comment 6**: p4 l5ff: Since the increases and decreases in elevation are artificial, to what degree are the simulated d18O anomalies applicable to real-world reconstuctions? The precipitation-weighted d18O will greatly depend on precipitation and therefore barrier effects (see above).

**Response to minor comment 6:** The 16 idealised simulations allows isolating the isotopic and climate response to GIS elevation changes. In addition, we have also 32 simulations with more (arguably) physical LIG GIS morphologies based on an initial ensemble of 14 LIG GIS reconstructions (Robinson et al., 2011; Born and Nisancioglu, 2012; Helsen et al., 2013; Quiquet et al., 2013; Stone et al., 2013; Calov et al., 2015) (see section 2.2.2). We then combine the results of the first (more idealised GIS) and second (more physical GIS) set of simulations to explore LIG lapse rates and $\delta^{18}O$ changes and compare them with observational estimates (Figures 7, 8 and 9). We consider that this is a sensible procedure, which makes our model results applicable and comparable to real-world reconstructions. In addition, we agree with the reviewer that $\delta^{18}O$ depends on precipitation changes and barrier effect; we have added new analysis of this factor (see response to major comment 3).

**Minor comment 7**: figure 1: Why use a piecewise linear fit? Nonlinearity is referred to in the text.

**Response to minor comment 7:** We perform a fit of the data using a piecewise linear function; this consists of two discrete linear segments which are used to describe the dependent variables; these are $\delta^{18}O$, temperature, precipitation and winter sea ice. Two different linear regression models are fitted to the data, one for increases in GIS elevation and another one

for decreases. We apply a piecewise linear fit to all variables because this makes (1) the paper easy to follow and understand and (2) it makes comparison of our results with previous studies easier. Following the reviewer's suggestion, we have added a second best fit of the data using an exponential function, which confirms the non-linearities of the climate response to GIS elevation changes.

Page 19, Fig. 1: "Additionally, a second fit of the data using an exponential function (y = ±ae$^{-bx}$ + c) is included (red lines with dashes)".

[Figure]

**Minor comment 8**: figure 2: Is there any sign of inversion in winter or other local temperature effects like in Merz et al.? How would this impact d18Op?

**Response to minor comment 8:** See response to major comment 3.

**Minor comment 9**: figure 4: Why not show anomalies like in all other figures?

**Response to minor comment 9:** In most of the manuscript we study/report changes in sea ice between the sensitivity simulations and the 125 ka control (e.g. Fig. 1, Fig. A6 and Table

A1). Because of this, we consider useful to show in Fig. 4 the absolute values of sea ice concentration for the 125 ka control.

**Minor comment 10**: p5 l10: typo: aN increase

**Response to minor comment 10:** Done.

**Minor comment 111**: p5 l15: This part would benefit from looking at the findings of Hakuba et al. (2012) and Merz et al. (2014a). Discuss why local dynamic and temperature changes are not seen in HadCM3 as compared to CCSM4.

**Response to minor comment 11:** See response to major comment 3.

**Minor comment 12**: p6 l6: "likely linked to the reduced winter sea ice" Isn't the 2nd ensemble there to test this hypothesis?

**Response to minor comment 12:** Changes in winter sea ice in the Baffin Bay are rather small, thus we were unsure of whether there was also another mechanism at play. It may be that the different behaviour found at Camp Century site is also linked to its coastal position. Slope areas receive more moist air masses which are orographically lifted and consequently condensate and precipitate. Please see also response to minor comment 9 (reviewer 1).

The objective of the set of 32 simulations is to study the joint impacts of sea ice change and GIS change. To examine the influence of merely GIS height changes, we use the set of 16 simulations with idealised variations in the elevation of the GIS, where the present-day GIS extent (land-ice mask) is unmodified. Similarly, the sea ice retreat simulations of Malmierca-Vallet et al. (2018), which only include sea ice forcing changes, help to isolate the influence of $\delta^{18}O$ due to sea ice variations. We acknowledge that, like all approaches, this approach also has limitations. The interaction among the three factors (GIS elevation and extent and sea ice) could lead to smaller/larger effects than predicted from the sum of single parameter effects. We have added a new appendix (Appendix B) that explores the robustness of this approach. See also response to minor comment 2 (reviewer 1).

**Minor comment 13**: p6 l21: Pausata reference is on glacial climate and a change in elevation of the Laurentide ice sheet. Not relevant here.

**Response to minor comment 13:** The Pausata reference has been removed.

**Minor comment 14**: p7 l8: "to isolate the effect of sea ice" How? Greenland topography changes here too.

**Response to minor comment 14:** The sea ice retreat simulations of Malmierca-Vallet et al. (2018), which only include sea ice forcing changes, help to isolate the influence of $\delta^{18}O$ due to sea ice variations. We acknowledge that this approach has limitations as the interaction among the three factors (GIS elevation and extent and sea ice) could lead to smaller/larger effects than predicted from the sum of single parameter effects. We have added a new appendix (Appendix B) that explores the robustness of this approach. See also response to minor comment 2 (reviewer 1).

**Minor comment 15**: figure7: To my eye, sea ice correction does not really improve the fit. Overall, the"GIS" ensemble does not show a clear correlation between d18Op and ice core

elevation. The cloud of points are rather round. Does the fit stem from all simulations or only the 1st ensemble? What would a fit (and std.dev.) for only the GIS ensemble look like? Is the sea ice correction straightforward? Does it not relate to the rather patchy (Fig. 6) precip anomalies? Is it not important in which region the sea ice reduction takes place?

**Response to minor comment 15:** The main objective of the sea ice correction is to test how sea ice changes may influence isotopic lapse rate values. All best fit curves in Fig. 7 stem for all simulations, not only the 1$^{st}$ ensemble. The figure below shows the best fit and Stdev for only the second set of simulations.

[Figure]

Sea-ice-corrected d18Op anomalies compared to 125Control. Dots represent results for the 32 simulations that examine the joined impact of Arctic sea ice retreat and modified GIS shape. Solid lines signify best fit curves (y = a+bx) and shade envelopes represent ±3s uncertainty on the best fit lines.

We have added a new appendix (Appendix B) that explores in more detail our sea ice correction method. See also response to minor comment 2 (reviewer 1).

We agree with the reviewer that $\delta^{18}O$ depends on precipitation changes and barrier effect; we have added new analysis of this factor (see response to major comment 3).

Our sea ice retreat experiments include a uniform retreat, a uniform sea ice forcing applied at the bottom of all Northern Hemisphere sea ice. If we were to examine the Greenland ice core $\delta^{18}O$ response to the spatial pattern of sea ice retreat, we will have to run, for example, a suite of experiments with an idealised spatially constrained sea ice retreat. In particular, the Arctic Ocean and/or North Atlantic could be split in several sectors and then forced sea ice retreat in each one of these sectors. We agree with the reviewer that this is a most interesting question. Answering it is however outside the scope of this work.

**Minor comment 16**: p7 l24: I might have missed the definition, but I do not understand what exactly "core average" means.

**Response to minor comment 16:** The following explanation has been added:

> Page 7, Line 6: (averaging across the six core sites: NEEM, NGRIP, GRIP, GISP2, DYE3 and Camp Century).

**Minor comment 17**: p7 l25: changes in gradient are described, followed by an argument that sea ice influences the linearity of the d18Op-elevation relationship. Gradients only describe the linear trend, so I am very confused over what this statement means. I think a non-linear metric, e.g., the curvature or similar, is needed.

**Response to minor comment 17:** Following the reviewer's suggestion, we have added a second best fit of the data using an exponential function, which confirms the non-linearities of the climate response to GIS elevation changes. See also response to Minor comment 7 and modified Fig. 1.

**Minor comment 18**: p7 l29: "The dependence of the δ18O variable on elevation variations occurs in response to variations in winter sea ice extent." This can be interpreted as if the authors want to claim that elevation changes are a result of sea ice anomalies, and that the latter are the ultimate reason for isotope changes. This touches on an important point: The relative importance of sea ice and elevation changes and feedbacks between the two effects, that should be discussed in much more details. I think the two ensembles are well suited for such an analysis.

**Response to minor comment 18:** The sentence linking $\delta^{18}O$ changes with elevation and sea ice variations has been modified and moved to section 3.1.5. With this sentence we claim that increases/decreases in elevation result in decrease/increases in sea ice, and ultimately this act as a positive/negative feedback on $\delta^{18}O$.

The authors have added two new sections with the titles "**Implications for NEEM $\delta^{18}O$ and elevation reconstructions**" and "**Relative influence of sea ice and GIS changes on ice core $\delta^{18}O$**". In this two sections, we: (1) compare the model results with the NEEM $\delta^{18}O$ and elevation reconstructions, and (2) we provide information about the relative importance of sea ice and ice sheet changes on determining $\delta^{18}O$ changes at each ice core site. See also response to minor comment 3 (reviewer 1) and major comment 2 (reviewer 2).

**Minor comment 19**: p8 l22: The combined changes discussed above are on the GIS ensemble, but figure 1 is on the elevation ensemble only. I think this is an error in the methodology.

**Response to minor comment 19:** The first part of the manuscript and consequently the first two sections of the discussion (sections 4.1 and 4.2) focus on the 1st ensemble of simulations. This helps examining the isolated impact of GIS elevation changes on Greenland $\delta^{18}O$ and underlying process (temperature, precipitation, arctic sea ice and atmospheric circulation). This is useful in that it helps improving our understating of the elevation signal captured in Greenland ice core records. The second part of the manuscript and consequently section 4.3 of the discussion focus on examining how sea ice changes may influence modelled isotopic lapse rate values over Greenland. For this analysis, we use the 32 simulations performed with more realistic sea ice retreat scenarios and GIS morphologies for the LIG period because: (1) this makes our analysis more accurate and, (2) this makes our isotopic lapse rates and $\delta^{18}O$ anomalies more comparable with observational estimates.  To make this point clear we have added the following text:

Page 2, Line 27: In this study, we first investigate the isolated impact of GIS elevation changes on Greenland $\delta^{18}O$ and the underlying processes (section 3.1). We perform a suite of idealised elevation change simulations with the isotope-enabled climate model HadCM3 to analyse the response of Greenland temperature (section 3.1.2) and precipitation (section 3.1.4), Arctic sea ice (section 3.1.5) and atmospheric circulation (section 3.1.3) to these GIS elevation changes. The second part of this study focus on testing to which extent variations in the background climate state (Arctic sea ice extent) may influence the isotopic lapse rate values at different Greenland ice core sites (section 3.2). For this analysis, we additional use a second set of simulations that investigate the joint impact of Arctic sea ice change and GIS changes.

**Minor comment 20**: section 4.3: This section compares modelled and observed isotopic lapse rates. It seems to me that the modelled lapse rates are derived from the first ensemble, i.e., it describes a change in d18Op as a function of the lower elevation at the same elevation, while the observed gradients have a spatial component, and thus a lower elevation generally implies a closer proximity to the coast. These should not be mixed or the difference must at least be clearly stated.

**Response to minor comment 20:** The modelled isotopic lapse rates are derived from the first and second ensemble of simulations. In addition, we report core-average lapse rates, averaging across the six ice core sites (NEEM, NGRIP, GRIP, GISP2, Camp Century and DYE3). As such our modelled lapse rates have also a spatial component. Following the reviewer's suggestion, we have added a definition for core-average lapse rate.

Page 7, Line 7: (averaging across the six core sites: NEEM, NGRIP, GRIP, GISP2, DYE3 and Camp Century).

**Minor comment 21**: There is an excessive use of appendix figures for a journal that does not have very strict length restrictions.

**Response to minor comment 21:**  We have removed fig C1 and F1 from supplementary information to the main text.

---

## Author Response (AR2)

Dear Editor,

Thank you for the letter and the reviewer's report. Please find below a point-by-point response to the reviewer's comments. A revised manuscript is attached with the main changes marked in blue.

We hope you find this satisfactory and look forward to hearing from you soon.

Yours sincerely,

Irene Malmierca Vallet, and co-authors.

**REVIEWER 2**

**Minor comments:**

**Minor comment 1:** p2 l3: "… believed to been have risen …", delete "been".

**Response to minor comment 1:** Done.

**Minor comment 2:** p2 l30: "The second part … focuses".

**Response to minor comment 2:** Done.

**Minor comment 3:** p7 l32: The reference on winds in glacial climate is still included (Pausata et al., 2011). Again, this is misleading because this study concerns a different background climate. Stone and Lunt, 2013 is not included in the list of references.

**Response to minor comment 3:** The reference of Pausata et al. (2011) has been removed. And, the reference of Stone and Lunt (2013) is now included in the list of references.

**Minor comment 4:** figure 1: I would have chosen a symmetric function for the nonlinear fit, such as $x^2$, unless there is a reason to expect an exponential dependency. However, there is no need to revise the figure.

**Response to minor comment 3:** For the nonlinear fit, we apply an exponential function because (1) it fits the data well and (2) it makes the comparison of results from different variables easier.